# Sustainable biomimetic solar distillation with edge crystallization for passive salt collection and zero brine discharge

Mohamed A. Abdelsalam[1,3], Muhammad Sajjad[1,3], Aikifa Raza [1], Faisal AlMarzooqi[2] & TieJun Zhang [1]✉

The urgency of addressing water scarcity and exponential population rise has necessitated the use of sustainable desalination for clean water production, while conventional thermal desalination processes consume fossil fuel with brine rejection. As a promising solution to sustainable solar thermal distillation, we report a scalable mangrove-mimicked device for direct solar vapor generation and passive salt collection without brine discharge. Capillarity-driven salty water supply and continuous vapor generation are ensured by anti-corrosion porous wicking stem and multi-layer leaves, which are made of low-cost superhydrophilic nanostructured titanium meshes. Precipitated salt at the leaf edge forms porous patch during daytime evaporation and get peeled by gravity during night when saline water rewets the leaves, and these salt patches can enhance vaporization by 1.6 times as indicated by our findings. The proposed solar vapor generator achieves a stable photothermal efficiency around 94% under one sun when treating synthetic seawater with a salinity of 3.5 wt.%. Under outdoor conditions, it can produce 2.2 L m$^{-2}$ of freshwater per day from real seawater, which is sufficient for individual drinking needs. This kind of biomimetic solar distillation devices have demonstrated great capability in clean water production and passive salt collection to tackle global water and environmental challenges.

With rapid economic development and population growth, over 30% of the human population is limited to no access to clean water, which has been recognized as the key sustainable development goal by the United Nations[1,2]. Electricity driven reverse osmosis and thermal desalination have been utilized for years to produce fresh water from seawater[3–5], but these technologies are energy intensive and directly or indirectly depend on fossil fuels with significant greenhouse gas emissions[6]. The brine usually rejected from conventional desalination plants to seawater has raised serious environmental concerns, because the increasing seawater salinity is harming the aquatic life. In addition, conventional desalination technologies heavily rely on fossil fuels with

substantial carbon footprint[7–12]. Therefore, there is pressing demand for clean and environment-friendly desalination technologies driven by renewable energy[13–16], among which direct solar vapor generation is a promising solution. Driven by proper thermal management and localized heating, interfacial solar vapor generation units produce clean water by harnessing solar energy[17–21]. Various structural designs and material advancements have been reported to fabricate efficient solar vapor generators[22–24]. Direct solar vapor generator can also be employed for treating brine solutions without liquid discharge, making the dry salt as the only byproduct[18,19,25]. However, along with the strong evaporation at the interface, salts in the bulk solutions inevitably tend

[1]Department of Mechanical and Nuclear Engineering, Khalifa University of Science and Technology, P.O. Box 127788 Abu Dhabi, United Arab Emirates. [2]Department of Chemical and Petroleum Engineering, Khalifa University of Science and Technology, P.O. Box 127788 Abu Dhabi, United Arab Emirates. [3]These authors contributed equally: Mohamed A. Abdelsalam, Muhammad Sajjad. ✉e-mail: tiejun.zhang@ku.ac.ae

to accumulate on the photothermal layer of the evaporator, hindering the light absorption and the vapor release that eventually decrease the overall thermal efficiency and the applicability of solar vapor generators. Current passive strategies to mitigate this serious drawback involve but not limited to, Janus (hydrophilic/hydrophobic) structures that allow water to evaporate while blocking salt crystallization[22], direct contact solar evaporators separating the brine solution from the evaporation surface and allowing the accumulated salt to diffuse back in dark[26], thus, maintaining a clean evaporation surface for continuous and stable performance. In most of these strategies, salt is diffused back to the water reservoir, consequently, increasing the salinity levels and wasting the opportunity for salt collection[10,17,22,23,26]. Therefore, besides producing fresh water, proper brine management is equally critical during any solar vapor generation process[27]. In this work, edge-preferred crystallization, which conforms with the concept of zero liquid discharge, has been utilized for the simultaneous production of salt and freshwater. Salt is directed and guided to crystalize on the evaporator edges while maintaining a clean central evaporating area for continuous operation.

Inspired by successful adaptation of natural halophyte plants, some research efforts have been made to adopt the biomimetic concept for various engineering applications, including fresh vapor production and salt management[24,28–31]. In particular, mangrove-inspired designs are promising candidates for designing bio-mimicked SVGs with salt regulating-mechanism[32,33]. Passive water transport and salt-secreting ability of mangrove trees have motivated us to fabricate a biomimetic solar-driven device, enabling simultaneous water distillation and salt collection without brine discharge[17]. Solar desalination unit, including a solar vapor generator and crystallizer (SVGC), is expected to have excellent light absorption, low flow resistance, scalability, and anti-corrosion characteristics[34]. Herein, we propose a foldable all-in-one mangrove-mimicked solar evaporator by using chemically etched titanium mesh, as shown in Fig. 1a, b. The nano/micro-structured titania layer on titanium mesh (TiO$_2$/Ti) surface

makes it an excellent solar absorber with superhydrophilicity and anti-corrosive properties. The evaporation and salt crystallization performance of the synthetic TiO$_2$/Ti leaves were systematically evaluated under different solar irradiances and brine concentrations. The day/night periodic operation was employed as a sustainable method for the co-production of freshwater and salts. Our characterization on salty water propagation reveals porous and patchy salt in clogged leaf-like structures can enhance the evaporation flux when compared with fully clean leaves. Furthermore, the factors affecting salt nucleation, growth, and peeling are investigated at the microscale. We demonstrate that the reduction in evaporation rates and thermal efficiencies of the salt-clogged solar evaporators is mainly due to light absorption losses rather than blocking the pores by salt, because the precipitated patchy salt contributes to higher evaporation rate under dark. This work paves a way for solar energy utilization, simultaneous freshwater production, and passive salt collection with zero brine discharge.

## Results

### Conceptual design of SVGC

Salt secretion and edge crystallization, intrinsically happening in natural mangroves with salt gland in leaves, is responsible for salt precipitation on the leaf surface (Fig. 1a). In our scenario, the biomimetic leaves have the ability to crystalize the salt on their edges, maintaining a clean central area of the leaf for continuous solar thermal evaporation and salt harvesting without performance deterioration. In this work, we have designed and developed an all-in-one mangrove-mimicked SVGC, where artificial leaves serve as the evaporation surface for vapor escape, and a hydrophilic stem continuously supplies water by capillary wicking, as shown in Fig. 1b. The top surface of the leaves harvest sunlight while acting as the evaporation site. When in contact with water, the stem of the foldable structure enables passive unidirectional water transport from the bulk water to the leaf surface owing to the capillary forces associated with the porous structure. Figure 1b inset shows the optical image of mangrove mimicked SVGC,

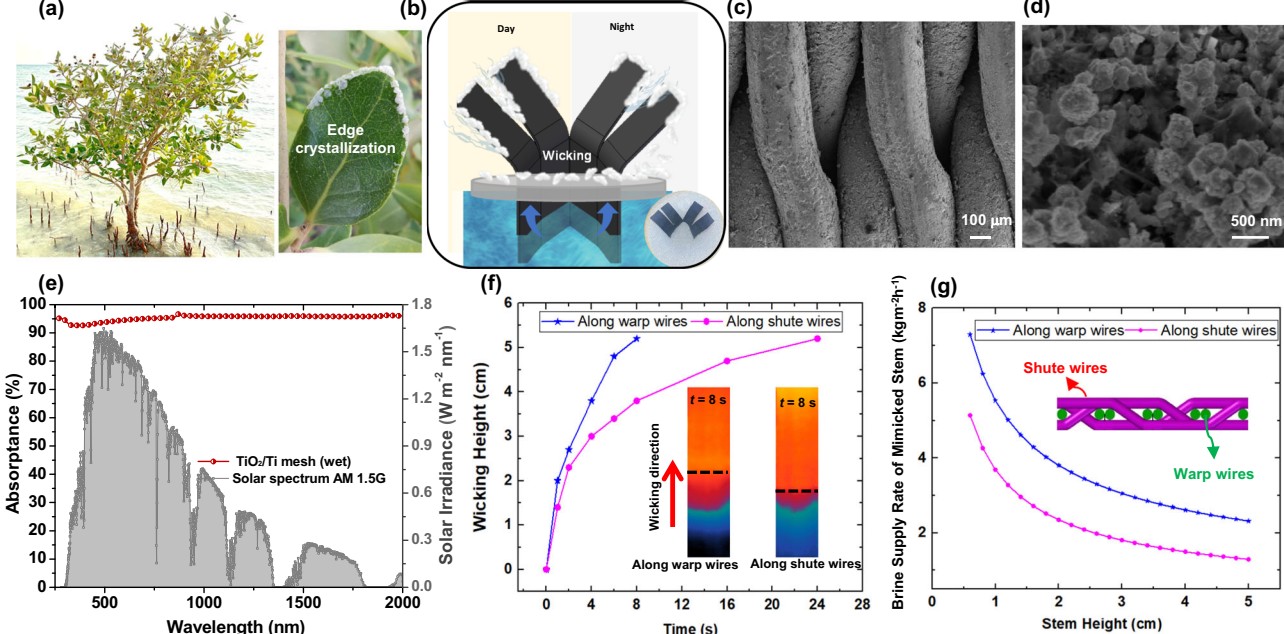

**Fig. 1 | Conceptual design of the mangrove-mimicked solar vapor generator and crystallizer (SVGC). a** Photo showing a grey mangrove (Avicennia marina) plant and a mangrove leaf with edge crystallization in Abu Dhabi, UAE. **b** Schematic showing design of the proposed mangrove-mimicked SVGC device and its working principle. SEM images of the oxidized mesh under (**c**) low and (**d**) high magnification. **e** Measured absorption spectra of the oxidized TiO$_2$/Ti mesh under the wet state in the wavelength range of 250−2000 nm. **f** Evolution of the wicking front of the saline water (24 wt.% NaCl) for the oxidized TiO$_2$/Ti mesh (data uncertainty: ± 0.8 mm). **g** Saline water (24 wt.% NaCl) lifting capacity of the mimicked stem (with a width of 40 mm and thickness of 0.89 mm equivalent to the proposed device) as a function of height.

which is made of nano/micro-structured titanium (TiO$_2$/Ti) mesh. Figure 1c shows the tight weaving patterns of TiO$_2$/Ti mesh wires captured by the scanning electron microscopy (SEM), with no observable gap between wires. The high-magnification SEM image in Fig. 1d indicates the dense coverage of nanostructured TiO$_2$ on titanium wires, which exhibit superior liquid propagation capability (detailed characterization is given in Supplementary Note 1). In addition, nanostructured TiO$_2$/Ti mesh is an excellent photothermal material that absorbs sunlight to boost the evaporation process without the need for extra light absorption coating. Once the superhydrophilic TiO$_2$/Ti mesh after oxidation gets saturated with water, it exhibits omnidirectional solar absorptance over 90%, as illustrated in Fig. 1e, in the ultraviolet, visible light and near-infrared wavelength range (250–2000 nm) (optical characterization given in Supplementary Fig. 4). By using infrared imaging[35,36], we investigated the imbibition of saline water (24 wt.% NaCl) along the nanostructured TiO$_2$/Ti mesh (artificial stem) in the biomimetic device by tracking the liquid front. The transient evolution of the waterfront along the mesh wires is shown in Fig. 1f, where the wicking height is 52 mm after 8 s of passive capillary pumping by the oxidized TiO$_2$/Ti mesh for wicking along warp wires. IR images of the liquid propagation in the mesh at $t = 8$ s are also depicted in Fig. 1f. The propagation of the liquid front (shown by the dashed line in Fig. 1f) is apparent in the IR images owing to the emissivity difference between the wetted and non-wetted regions of the mesh. It can also be observed that the IR image color for the wetted mesh part varies due to the varying thickness of liquid films along the propagation direction[37]. Additionally, from the velocity of the water propagation, we were able to determine the saline water supply rate as

a function of stem height (Fig. 1g). Our results have shown that the water supply rate is lower in the shute wires when compared to the warp wires. The pumping capacity is found to be significantly affected by the height of the stem, and it decreases as the stem height increases.

## Design optimization and performance analysis of SVGC

Figure 2a shows schematic of the experimental setup in our lab to characterize the SVGC (details given in methods section). The performance of a single-leaf SVGC was evaluated under simulated sunlight on the top surface, by varying the tilt angle of the leaf to the stem at −30, 0, and +30° but with the same projected illumination area. In all cases, brine with 12 wt.% salinity and irradiance of 1 kWm$^{-2}$ for 12 h were employed (detailed results and explanation given in Supplementary Note 2–4). In summary, the solar vapor generator with single leaf and a tilt angle of +30° indicated a substantial salt resistance ability and higher thermal efficiency of 75% compared to 60% and 62% for the leaf with 0 and −30° tilt angles, respectively. Furthermore, it also shows high potential for salt harvesting, therefore, we used the tilt angle of +30° in mangrove-mimicked SVGC in all the subsequent experiments.

The stem height is also a crucial parameter that affects our mangrove-mimicked SVGC's evaporation performance. Thus, a systematic comparison between two setups (with the same projected area) by controlling the height of the stem, since it was submerged in water, was conducted. The salinity of the bulk water solution was maintained at 3.5 wt.% and the experiments were conducted under 1 kWm$^{-2}$ for a period of 24 h while simulating the day and night alternation (see Fig. 2e and f). Our results have indicated that the short mesh is at optimal stem height to maintain strong liquid wicking at

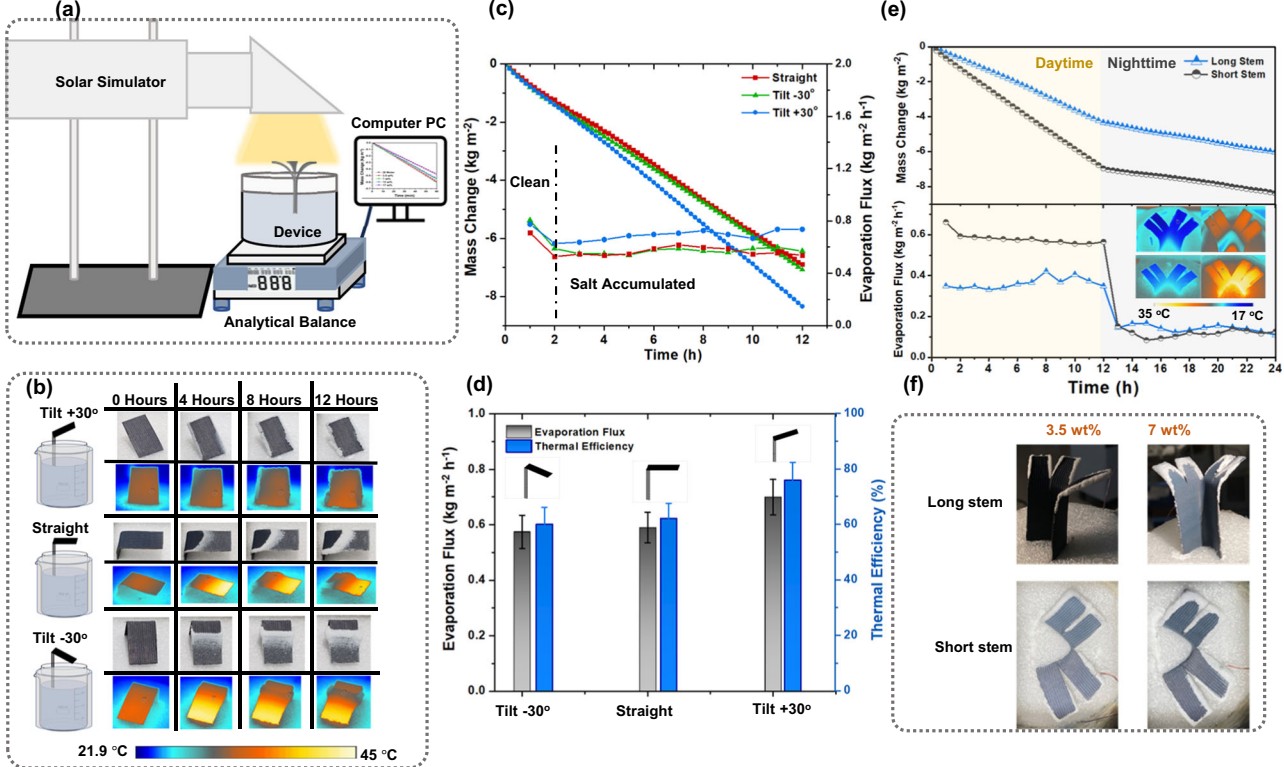

**Fig. 2 | Performance analysis of the proposed solar vapor generation-crystallization devices with varying leaf orientation and stem height.**
**a** Schematic of the experimental setup used to evaluate the lab scale device.
**b** Optical and IR images of three single-leaf evaporators with tilt angles of −30, 0 and +30°, all using saline water with salinity of 12 wt.% under one sun illumination.
**c** Comparison of single leaf solar vapor generator with various tilt angles in terms of mass change and evaporation flux per hour. **d** Average evaporation flux and

thermal efficiency for single leaf solar vapor generator with various tilt angles. Error bars represent the standard deviation of three-time measurements. **e** Comparison between long and short stem in mangrove-mimicked solar vapor generators in terms of mass change and evaporation flux for 24 h using water with salinity of 3.5 wt.%. **f** Photos of the mangrove-like solar vapor generator with long and short stem using water with salinities of 3.5 and 7 wt.% after 12 h of illumination.

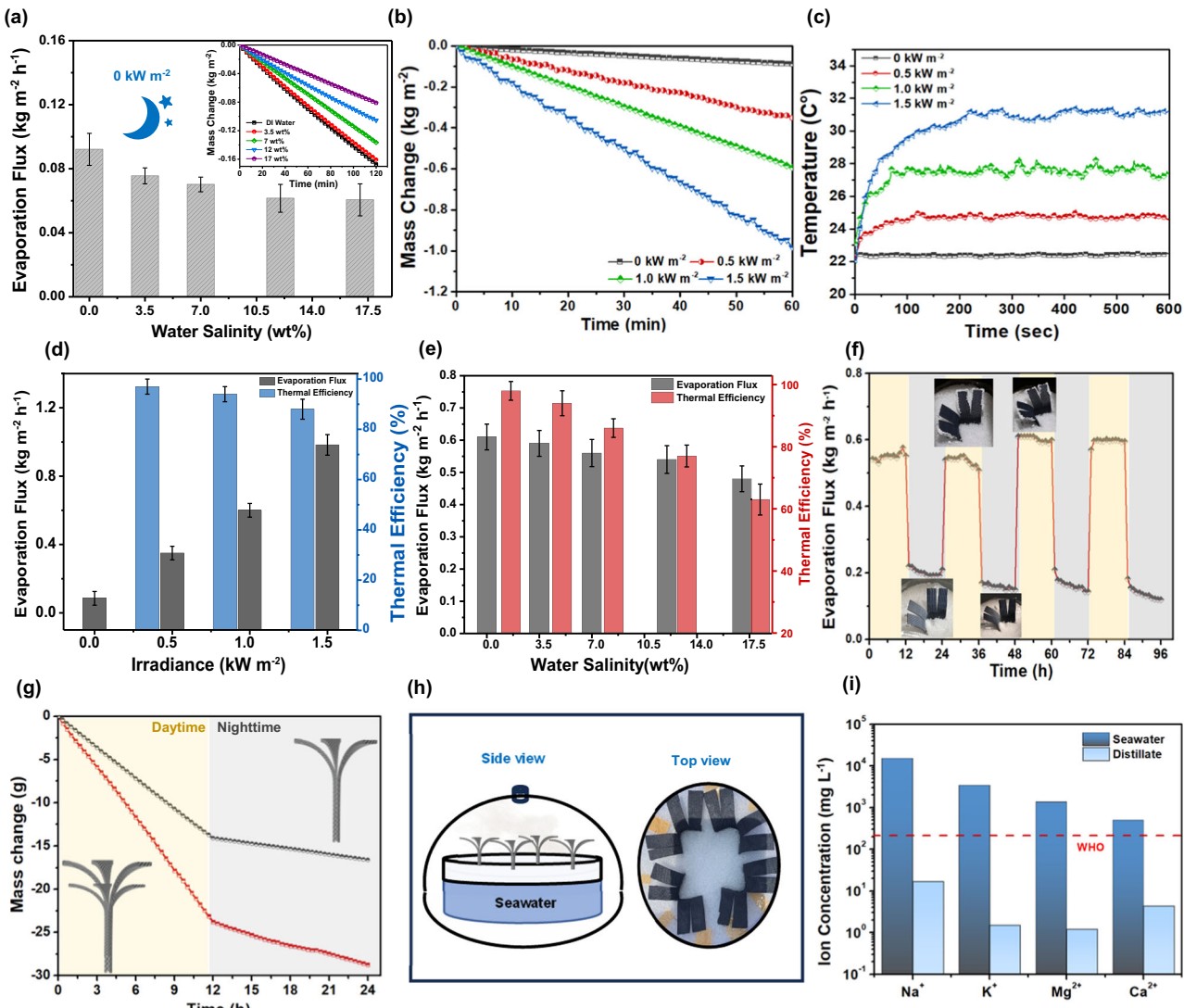

**Fig. 3 | Performance evaluation of the mangrove-mimicked solar vapor generator and crystallizer (SVGC) at different irradiances using water with different salinities. a** Dark evaporation fluxes and water mass changes (inset figure) for the mangrove-mimicked SVGC using water with different salinities (**b**) Water mass changes of the proposed SVGC under various intensities of light illumination (0, 0.5, 1, 1.5 suns). **c** Measured temperatures at the evaporation surface as a function of different intensities of light illumination (0, 0.5, 1, 1.5 suns). Average thermal efficiencies and evaporation fluxes at (**d**) different irradiances using 3.5 wt.% solution and (**e**) different salinities under 1 sun. Error bars represent the standard deviation of three-time measurements. **f** Durability test for the SVGC device for a period of 4 consecutive days using the solution with a salinity of 3.5 wt.%. **g**, **h** Scalable SVGC devices: (**g**) Performance comparison between single- and double-layered biomimetic SVGC devices in terms of mass change of water with salinity of 3.5 wt.%, (**h**) Outdoor experimental setup for freshwater collection consisting of four mangrove-mimicked SVGC. **i** Ionic concentration for the real seawater before and after solar thermal distillation.

different salinities for continuous solar vapor generation and salt collection (detailed results in Supplementary Note 4).

To assess the evaporation performance, a series of experiments were conducted by using water with different salinities, 0 wt.% (DI water), 3.5, 7, 12, and 17 wt.%. The evaporation performance of mangrove-like SVGC device under a dark environment is depicted in Fig. 3a. Our results indicate that the evaporation flux of the proposed biomimetic SVGC device with DI water was the highest with a value of 0.09 kg m$^{-2}$ h$^{-1}$ followed by 0.076, 0.072, 0.062, and 0.060 kg m$^{-2}$ h$^{-1}$ when water with salinity of 3.5, 7, 12, and 17 wt.%, respectively were used. In order to evaluate the effect of solar irradiance on evaporation performance, a series of solar vapor generation experiments were performed with SVGC by using simulated sunlight intensities of 0.5, 1, and 1.5 sun, respectively. The corresponding water mass change results are presented in Fig. 3b, and the recorded evaporation flux was 0.38 kg m$^{-2}$ h$^{-1}$ under 0.5 sun and increased to 0.61 and 0.98 kg m$^{-2}$ h$^{-1}$ for solar irradiances of 1.0 and 1.5 sun, respectively. However, the calculated

evaporation flux based only on the top solar illuminated area was 0.76, 1.22, and 1.92 kg m$^{-2}$ h$^{-1}$ for solar irradiances of 0.5, 1.0 and 1.5 sun, respectively. The temperature distribution given in Supplementary Note 13, confirms that the harvested solar thermal energy is localized and uniform at the top surfaces for all biomimetic leaves. The change in top surface temperature under different light intensities is shown in Fig. 3c. After being exposed to the simulated sunlight in lab conditions, the top surface temperature increased and then reached the steady-state conditions. It can be clearly seen from Fig. 3c that the top surface temperature was low under 0 sun with a value of 22 °C as the ambient. The top surface temperature has the same trend under various light intensities, and its steady-state value increases from ~25 °C at 0.5 sun to 28 °C at 1.0 sun and to 32 °C at 1.5 sun, respectively. In addition, as depicted in Fig. 3d, the variation in solar-to-vapor generation efficiency shows opposite trend to the evaporation flux, the efficiency first was 97% at 0.5 suns and then decreases to 94 and 88% (as the temperature of the leaves is increasing) for the 1 and 1.5 sun, respectively as shown in Fig. 3d.

Similar to the dark experiments, this SVGC device can produce water from artificial salt solutions with different salinities (0, 3.5, 7, 12, and 17 wt.%) under one sun (1 kW m$^{-2}$) illumination (Fig. 3e). The water mass change per unit area decreases with increasing concentration of the bulk water from 0 to 17 wt.%, because increasing ions in the solution tend to lower the free energy of the water molecules. The saturation vapor pressure of saline water would decrease with increasing salinity, eventually reducing the evaporation rate[38]. The mangrove-like SVGC enabled stable mass fluxes for different concentrations of saline water: the highest evaporation flux observed for DI water with 0.62 kg m$^{-2}$ h$^{-1}$, followed by 0.6 kg m$^{-2}$ h$^{-1}$ for the water with salinity of 3.5 wt.%, and 0.56, 0.54, and 0.48 kg m$^{-2}$ h$^{-1}$ for the water with salinity of 7, 12, and 17 wt.%, respectively. (1.24, 1.20, 1.12, 1.08 and 0.96 kg m$^{-2}$ h$^{-1}$ if the evaporation flux was calculated based on the top solar area only). Our results showed that the thermal efficiency was high when compared with other SVG devices[26,39–42] ~ 98% for the deionized water, while it was 94, 86, 77 and 63% for the water with salinity of 3.5, 7, 12, and 17% wt.% solutions, respectively. The gradual reduction of evaporation efficiency occurred due to the decrease in the vapor pressure as the salt concentration in the solution increases (Fig. 3e). On the other hand, latent heat of vaporization of the salt solutions is lower than DI water (2440 kJ kg$^{-1}$ for deionized water and 1934 kJ kg$^{-1}$ for the solution with salinity of 17 wt.%, which may result in lower thermal efficiency owing to increased surface temperature[43].

To ensure the durability of the SVGC device, cyclic experiments using water with salinity 3.5 wt.% were carried out through four consecutive days, under 1-sun illumination for 12 h and under dark environment for 12 h. Figure 3f shows stable evaporation fluxes of 0.6 and 0.2 kg m$^{-2}$ h$^{-1}$(based on total and top areas) over all the simulated day and night cycles, respectively. The detailed results for indoor and outdoor solar vapor generation experiments are discussed in Supplementary Note 6. In addition, by fully utilizing the gaps among the SVGC leaves, we fabricated a larger tree-like device with two layers of leaves. Compared with the single-layered device, the double-layer tree-like SVGC device exhibited better evaporation performance (Fig. 3g) under the same solar beam area, demonstrating the compactness of our SVGC.

Furthermore, in order to assess the overall distillation performance of the mangrove-like SVGC, actual seawater with measured salinity of 4.2 wt.% obtained from Al Hudyriat Island (Abu Dhabi) was utilized for freshwater collection (further details in Supplementary Note 5). As the demonstration of the proposed evaporator for scalable applications, four biomimetic devices were used in the experiments to amplify the total evaporation area as shown in Fig. 3h. The system was able to produce daily fresh water with a flux of 2.2 L/m$^2$ day, which is enough to satisfy daily individual drinking needs. Besides, we used the inductive coupled plasma test to measure the ionic concentration before and after distillation. As depicted in Fig. 3i, the concentration of the various cations (i.e, sodium, magnesium, potassium, and calcium) in distillate is reduced by 4 orders of magnitude, when compared to real seawater. The concentrations of the distilled water (light blue columns in Fig. 3i) are 16.84 mg L$^{-1}$ for Na$^+$, 1.49 mg L$^{-1}$ for K$^+$,1.15 mg L$^{-1}$ for Mg$^{2+}$, 4.29 mg L$^{-1}$ for Ca$^{2+}$, respectively. The ion concentrations in the collected water satisfy the World Health Organization[44] and Oman Humanitarian Desalination Challenge (OHDC)[45] standards for drinking water, which confirms the effectiveness of our SVGC device for practical applications (Supplementary Note 8).

### Periodic operation of mangrove-like SVGC

To evaluate the co-productivity of salt and freshwater for the mangrove-mimicked SVGC, a set of continuous experiments were conducted in controlled lab environment for consecutive 24 h under 1 kW m$^{-2}$ irradiance as shown in Fig. 4a. The proposed SVGC device was placed for 12 h under direct solar irradiance followed by 12 h under a dark environment, similar to the natural outdoor environment. The evaporation rate when the light was on remains stable as demonstrated by the linear water mass change (Fig. 4a). The reason for stable evaporation flux is that salt crystals were noticeably growing on the edges of the leaves while the central area remained clean for light absorption. However, while using high-concentration solutions with a salinity of 12 wt.% and 17 wt.%, it was observed that salt accumulation occurred in the middle area of the leaves of plant, as shown in Fig. 4b. It is noteworthy that when the light was turned off during the night cycle, the evaporation flux significantly decreased to around 0.1 kg m$^{-2}$ h$^{-1}$ for four different salinities. Besides, it was observed that the thick salt layer on the edges of the plant-shaped structure was self-defoliated and passively peeled off owing to the dissolution and back diffusion induced by strong water rewetting. Eventually, the peeling led to the recovery of clean leaves, indicating that the plant-shaped structure is highly stable and reusable without significant variation in the evaporation performance as presented in Fig. 4b. The peeled salt mass percentages for different salinities were 100% for the 3.5 wt.%, followed by 100, 80, 0% for the 7, 12, and 17 wt.% saline water, respectively. Salt crystallization and accumulation during the day (while light is on) and passive salt peeling during the night (while light is off) ensure that the device can operate continuously for extended periods of time, (as shown in the Supplementary Movie 1 and indoor and outdoor reliability tests in Supplementary Note 6). Figure 4c summarizes the results of the co-production of salt and freshwater using saline water of different salinities. It can be noticed that the evaporator with 3.5 wt.% concentration produced salt with a peeling rate of 7.5 g m$^{-2}$ h$^{-1}$ with a corresponding highest average evaporation flux of 0.48 kg m$^{-2}$ h$^{-1}$ (yellow column in Fig. 4c). Similarly, the 7 wt.% solution generated salt with a rate of 11.6 g m$^{-2}$ h$^{-1}$ and an average evaporation flux of 0.36 kg m$^{-2}$ h$^{-1}$. The peeling rate was 33.2 g m$^{-2}$ h$^{-1}$ for 12 wt.% solution and the corresponding evaporation flux was 0.32 kg m$^{-2}$ h$^{-1}$. For the case of 17 wt.% solution, the reported evaporation flux was 0.26 kg m$^{-2}$ h$^{-1}$ with negligible salt peeling.

Surprisingly, during the dark cycle, the evaporation rate of the saline water with a concentration of 17 wt.% (~0.094 kg m$^{-2}$ h$^{-1}$) is even higher than that for 7 wt.% and 12 wt.% solutions as shown in the black columns in Fig. 4c. This is because the crystalized salt is porous and hydrophilic, with enhanced evaporation area owing to extended surfaces of the precipitated salt. However, when exposed to light, white salt crystals tend to reflect the sunlight and hinder the photothermal energy conversion. Thus, the reduction of evaporation rates and thermal efficiencies of the clogged solar evaporators is due to the light reflection instead of the pore blocking since the precipitated salt actually contributed to higher evaporation rate under dark as discussed above. Furthermore, to precisely investigate the hydrophilic property of the crystalized salt, the evaporation performance of clean and used devices (with patchy salt) was compared. The used device was fully covered with salt (5 g) with an average salt thickness of 0.4 cm, and solution with high salt concentration (17 wt.%) was employed to minimize the dissolution of the patchy salt for the used device (Fig. 4d). As the experiments were carried out in an absolute dark environment, there was no light reflection by the white salt. The evaporation rate of the stem with patchy salt was found to be 0.16 g h$^{-1}$, 1.6 times the evaporation rate of clean stem (i.e., 0.1 g h$^{-1}$), which confirms that patchy salt enhances the evaporation rate in dark environment.

In order to study the variation in salt concentration along the stem of SVGC, we carried out the finite element method (FEM) simulation using COMSOL V5.6 to characterize the spatial salt distribution along porous evaporator. The computational domain for numerical simulation is shown in Fig. 4e (left image). The simulation methodology and detailed results are reported in Supplementary Note 10. The experimental and simulation results for salt concentration along the evaporator length for 12 wt.% saline water are shown in Fig. 4e (middle and right, respectively). It can be seen that the salt concentration increases along evaporator length and reaches the saturation value, $c_{sat}$, (i.e.,

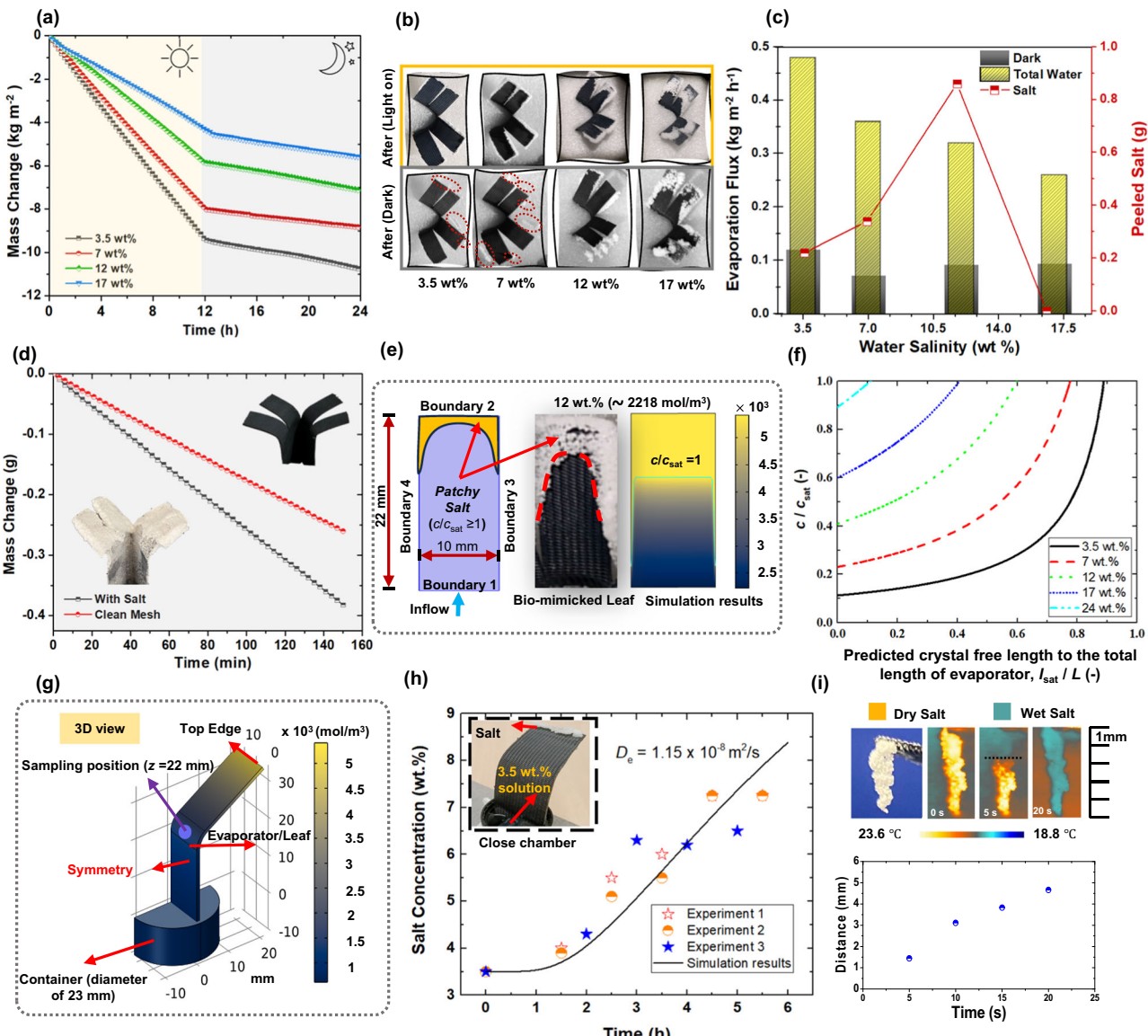

**Fig. 4 | Periodic operation of the proposed solar vapor generator and crystallizer (SVGC). a** Water mass changes for full-day evaporation experiments during the day and night operation. **b** Optical images for mangrove-mimicked SVGC after one day cycle (yellow box) and after one-night cycle (grey box) using water of various salinities. **c** Summary of the water vapor and salt harvesting coproduction during day and night. **d** Comparison between clean and salty evaporation devices in terms of mass change. **e** The computational domain for COMSOL V5.6 simulation (left subplot) to study the effect of salinity on salt concentration profile along the evaporator length. Middle and right subplots in (**e**) show experimental and simulated steady-state concentration profiles for 12 wt.% saline water, respectively. The

green line/curve represents $c/c_{sat} = 1$ (right subplot). **f** Variation in precipitation-free stem length to the total evaporation length ($l_{sat} / L$) as a function of dimensionless salt concentration ($c/c_{sat}$). **g** Computational domain for COMSOL V5.6 simulation to estimate the back diffusion coefficient using single-leaf evaporator and its volumetric concentration profile at $t = 6$ h in mol/m³, (**h**) Variation in salt concentration (wt.%) over time at $z = 22$ mm (solid black line for simulation, scatter points for experiments). The inset image of Fig. 4h shows experimental setup to study back diffusion. **i** IR imaging of the liquid propagation through patchy salt and the corresponding wicking distance over time.

26.3 wt.% or ~5411 mol/m³) at a certain distance from the inlet (i.e., Boundary 1). As the bulk (or inflow) concentration increases, the length of the probable crystal-free region is reduced. This is applicable to the cases with and without edge effect. Figure 4f shows the variation in dimensionless salt concentration ($c/c_{sat}$) as a function of precipitation-free stem length to the total evaporation length ($l_{sat} / L$). When there is no edge effect, the predicted $l_{sat} / L$ values corresponding to $c/c_{sat} = 1$ for 3.5, 6, 12, 18, and 24 wt.% are 0.89, 0.77, 0.6, 0.40, and 0.11, respectively. These simulation results in Fig. 4e-f are consistent with the experimental results for salt precipitation with our bio-mimicked SVGC in Fig. 4b.

To assess the back diffusion coefficient, the experimental setup along with the simulation domain are shown in Fig. 4g, h. Figure 4g

illustrates the volumetric contour profile after 6 h of back diffusion, while Fig. 4h gives experimental and simulation results for the variation in salt concentration over time at $z = 22$ mm. The diffusion coefficient is found to be $\sim 1.15 \times 10^{-8}$ m²/s when the simulated concentration profile matches the experimental data. The magnitude of the diffusion coefficient for our device is one order higher than that for bulk solution ($\sim 1.5 \times 10^{-9}$ m²/s). The simulation methodology and detailed discussion for the estimation of back diffusion coefficient are given in Supplementary Note 11. With the help of the IR imaging, we also characterized the liquid propagation through porous and patchy salt, where the patchy salt with a length of 5 mm precipitated at the edge of the stem (Fig. 4i) was used for this purpose. Time-lapse IR images with an interval of 5 s in the salt rewetting process are shown in

Fig. 4i. The evolution of the liquid front is distinguishable in the IR ranges, and the wetted salt is represented by blue while the dry salt is represented by orange color. In addition, wicking distance as a function of time is also reported in the bottom subplot of Fig. 4i (for details please see Supplementary Note 12).

## Discussion

The precipitation behaviour depends on the distribution of salt concentration on porous evaporator surface, wires weaving pattern and nanostructures on individual wires. Based on the classical nucleation theory, the impact of these parameters on energy barrier for heterogeneous nucleation ($\Delta G_{het}$) can be expressed as[46]:

$$\Delta G_{het} = \left(-4\pi r^3 \rho_s \Delta\mu/3 + \pi r^2 \gamma_{lc}\right)\Delta G^* \tag{1}$$

where $r$ is the radius of the spherical nucleus of new phase (i.e., crystals), $\rho_s$ is number density of crystals, $\Delta\mu$ is the difference in chemical potential of solute (i.e., NaCl salt) in solution between supersaturated and saturated states. $\gamma_{lc}$ is the interfacial tension between liquid phase (i.e., solution) and crystal, while $\Delta G^*$ is the the ratio of heterogeneous to homogeneous nucleation energy barrier which depends on pore structure and interfacial properties of new phase and saline solution (reported in detail in Supplementary Note 10). Under low super saturation at constant temperature and pressure, the chemical potential difference for evaporative crystallization of individual species is given by[47]:

$$\Delta\mu = RT\left(c/c_{eq} - 1\right) \tag{2}$$

where $R$ is the universal gas constant and $T$ is the absolute temperature. Similarly, $c$ and $c_{eq}$ are actual and equilibrium (or saturated) salt concentration in solution, respectively. Based on COMSOL Multiphysics V5.6 simulation, we predicted the probable location where concentrations could reach supersaturation (as reported in Fig. 4e, f). An increase in NaCl concentration will enhance the chemical potential difference of salt in the solution (Eq. 2), thus reducing the energy barrier for salt nucleation (Eq. 1). The cavities and nano/micro-structure of the SVGC device also reduces energy barrier as discussed in detail in Supplementary Note 10. The overall effect of the low energy barrier, which facilitates nucleation/crystallization, will be the same at all locations of the evaporator, while observed crystallization is not uniform over the evaporator surface. This clearly demonstrates the dominant impact of chemical potential difference of salt on crystallization, compared with the reduction in nucleation energy barrier due to micro-structure and cavities (quantified through $\theta$ and $\psi$ (Fig. 5) as discussed in Supplementary Note 10). Moreover, the precipitation behavior observed in experiments is in consistency with the concentration profiles obtained through COMSOL V5.6 simulation, which confirms the important role of supersaturation in crystallization (Fig. 4b, e, f).

Like the main evaporator surface (where we have $l_{sat}/L$ defined by $c/c_{sat} = 1$ as given in Fig. 4f), concentration will also increase along the spike's length and saline water at the tip of spikes will reach the supersaturation state earlier than that at the spike base (Fig. 5a, b). Therefore, higher supersaturation at the spike tip will increase chemical potential difference ($\Delta\mu$) of salt, thus reducing nucleation barrier. However, crystallization behavior will not be same on all spikes, because it also depends on the distance of a particular spike from the evaporator inlet, which affects salt concentration at the base of the spike. Spikes far from the evaporator inlet will have high concentration at their base, so these spikes will have more salt crystallization in comparison with the ones near the evaporator inlet (Fig. 5a). In addition, the evaporation rate around the spikes is higher than that on the main evaporator surface since it is easier for vapor to diffuse into air near spikes. Moreover, the liquid film at/between the spikes could

be thinner than that at the main surface, and higher evaporation at thin film region will lead saline water to reach supersaturation ($c/c_{sat}$) more easily, hence more crystallization. The dissolution of precipitated salt is more effective at the non-spike edge (Edge-2, Fig. 5f, h) owing to the direct contact between salt and the main leaf, in comparison with thee case with spikes (Edge-1, Fig. 5e, g), where the salt remains hung over the spikes (Fig. 5b, c–f). Once the first few micrometers' layers of the salt are dissolved back, the salt will be peeled off under gravity as shown in Fig. 5h. Therefore, spikes are not desirable from a salt-peeling perspective; however, they favor salt crystallization.

To sum up, edge-preferred crystallization, complying with the concept of zero brine discharge, has been employed for simultaneous production of salt and freshwater. Our scalable mangrove-mimicked passive device has achieved direct solar vapor generation and zero brine discharge simultaneously, showing a high solar thermal efficiency of 94% under one-sun irradiance owing to the omnidirectional superior light absorption, stability, and wicking capability. Anti-corrosion porous wicking stem and leaves made of nanostructured $TiO_2$/Ti meshes supply salty water via capillary pumping and enable continuous vapor generation until solid salt precipitates at leave edges. Our experiments have indicated that the patchy salt in clogged leaves-like structures can enhance the evaporation compared with fully clean leaves. Salt peeling can occur passively along artificial leaves during night, because the precipitated salt patch from leave edge can be rewetted by saline water to detach and fall under gravity, leading to passive salt collection. In addition, the proposed mangrove-like SVGC device can have multiple layers and branches volumetrically, which can harvest sunlight with any incident angle. Moreover, multiple mangrove-like devices can be placed in parallel to achieve scalable freshwater production with limited land use. In summary, this work has provided a sustainable solution for clean water production and brine treatment with abundant solar energy.

## Methods

### Material processing and characterization

To create the SVGC, the titanium mesh was first engineered through the oxidation process to enhance wettability and light absorption. After cleaning, titanium mesh was chemically etched with 1.00 molar aqueous solution of Sodium Hydroxide (Sigma-Aldrich, 99.9% purity) through the hydrothermal method to grow nanostructured $TiO_2$. (Supplementary Fig. 1). The morphology and microstructure of mesh were characterized by the scanning electron microscopies (Nova Nano SEM 650). The light absorption spectra of samples were recorded by UV-vis-NIR spectrophotometer (PerkinElmer, LAMBDA 1050). For the water propagation experiment within the mesh, the waterfront is identified by infrared (IR) imaging via the IR camera (FLIR E95). The contact angle tests were carried out by using a microscopic goniometer (Kyowa DM501). Detailed fabrication methods and characterization of nanostructured titanium meshes are added to Supplementary Note 1.

### Fabrication of lab-scale mangrove like SVGC

To test the SVGC under simulated sunlight, we built up a setup where a lightweight plastic beaker was utilized as a bulk water container and a thermal insulator from foam (thermal conductivity of ~0.03 W/mK) constructed on the top and the sides of the beaker. The experimental setup consists of a solar simulator (Sciencetech, Canada) with adjustable light intensity and an analytical balance (RADWAG) with an accuracy of 0.01 g. Along with DAQ (from National Instruments, NI MAX), thermocouples were utilized to measure and record the temperature of the light absorption surface (leaves). The solar intensity is measured by solar meter from EKO.

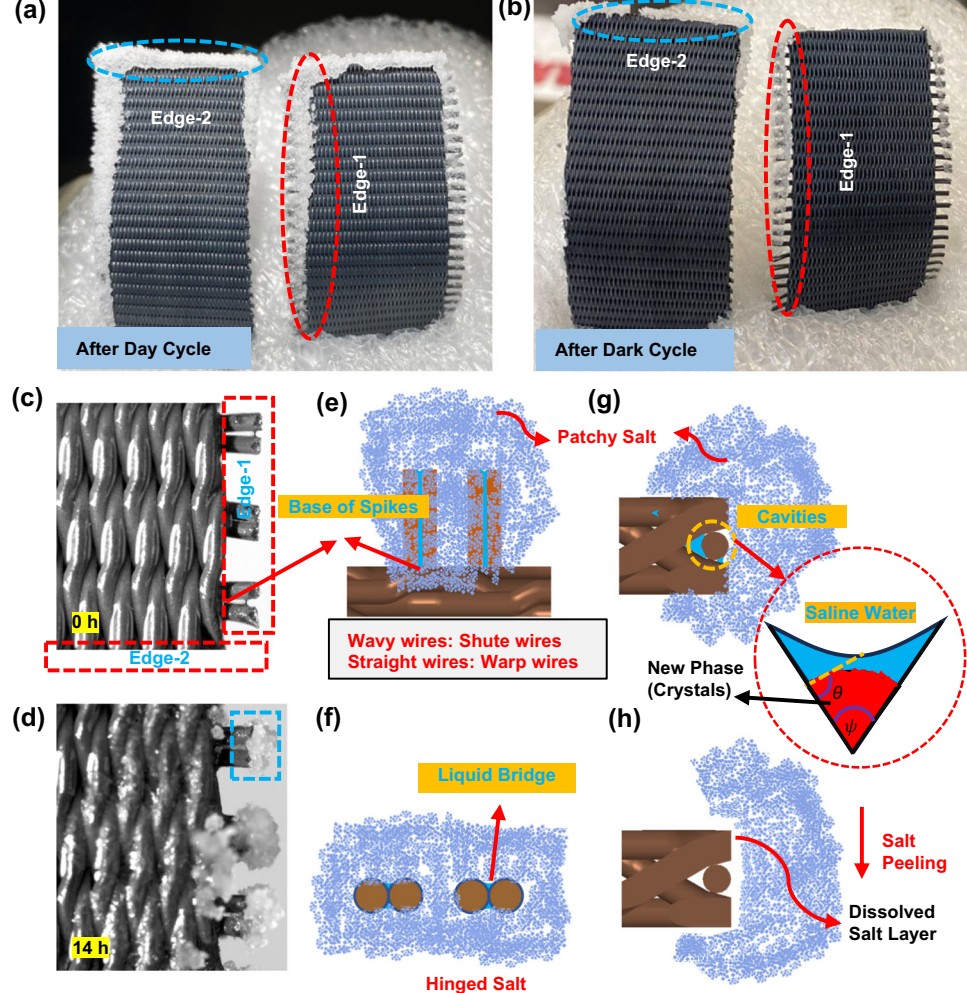

**Fig. 5 | Salt precipitation and peeling mechanism for the leaf (i.e., mesh) with and without edge spikes. a**, **b** Optical images showing precipitated salt at the edges of mesh with and without extended spikes after one day and night cycle, respectively for 7 wt.% saline water. **c**, **d** High magnification optical images showing salt precipitation on extended warp wires of leaf at (**c**) $t = 0$ h and (**d**) $t = 14$ h under ambient conditions. Illustration for the salt precipitation on spikes of the warp wires from (**e**) top and (**f**) side views. Schematic for the growth and peeling of salt at non-spike edge of the leaf/mesh from (**g**) top and (**h**) side views. **g** Also illustrates the nucleation of new phase (i.e., salt crystals) inside the cone-shaped cavity filled with saline water. The dimensions of the extended warp wires and shute wires are exaggerated for clarity in the schematic given in (**e**–**h**).

## Data availability

The data that supports the findings of the study are included in the main text and supplementary information files. Raw data files are available from the corresponding author upon request.

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

## Acknowledgements

This work was supported by the Abu Dhabi Award for Research Excellence 2019 (#AARE19-185, #AARE19-067) of ASPIRE under the Advanced Technology Research Council in Abu Dhabi UAE [T.J.Z., F.A.M.] and also by Sandooq Al Watan Applied Research & Development Grant (Project# SWARD-S19-003) [F.A.M., T.J.Z.]. The authors thank Prof. Shaojun Yuan at Sichuan University for providing raw titanium meshes and Dr. Maguy Abi Jaoude, Dr. Hongxia Li at Khalifa University for early discussions.

## Author contributions

T.J.Z. and M.A.A. proposed the concept and setup. M.A.A. and M.S. developed the set-up and conducted various experiments. A.R. helped in the characterization of structured titanium meshes. M.S. developed the numerical model and thermodynamic analysis for salty water transport and precipitation. M.A.A., M.S., A.R., F.A., and T.J.Z. all contributed to writing and revising the paper.

## Competing interests

The authors declare the following competing interests: T.J.Z., M.A.A., M.S., and F.A.M. have filed a US patent application based on the results of this work. The remaining authors declare no competing interests.
