## [Peer Review File · Nature Communications]

Sustainable Biomimetic Solar Distillation with Edge Crystallization for Passive Salt Collection and Zero Brine DischargeREVIEWER COMMENTS

Reviewer #1 (Remarks to the Author):

In this manuscript, the authors introduce a device that integrates a solar vapor generator with a salt crystallizer (SVGC) utilizing edge-preferred crystallization. This device aims to facilitate clean water production and passive salt collection. The authors' findings indicate that the presence of porous and patchy salt within structures resembling clogged leaves can notably enhance the evaporation flux, as compared to completely clean surfaces. Additionally, the authors observe that the decline in evaporation rates and thermal efficiencies of salt-clogged solar evaporators primarily result from losses due to light reflection, rather than pore blockage caused by salt accumulation.

In my view, this manuscript merits acceptance for publication, contingent upon the comprehensive addressing of the following comments:

- A lucid theoretical exposition of edge-preferred crystallization, ideally rooted in thermodynamics, is currently absent. I strongly encourage the authors to delve into this aspect. Particularly, it is crucial for the authors to expound on the circumstances (pertaining to textile structure, border end-points, etc.) under which crystallization is more prone to occur at the edges.
- The fabrication of the proposed textile involves several stages: titanium mesh, oxidation process, and chemical etching. A more intricate characterization of each stage is desirable. For instance, how does the material color evolve during fabrication—from pure titanium to titanium oxide and eventually to nanostructured titanium oxide? Additionally, when referring to titanium oxide, it would be valuable to specify the relevant phase (cubic, anatase, rutile).
- The authors allude to the occurrence of back diffusion during nocturnal/dark operation. Have the authors conducted an estimation of the back diffusion coefficient? Does its order of magnitude align with standard self-diffusion values? Alternatively, should a Marangoni effect be taken into consideration? (For reference, see <https://doi.org/10.1039/D0EE01440K>).
- The authors assert that the proposed material is foldable. I find it somewhat unclear how this foldability offers substantial advantages and ensures optimal light absorption.
- Lastly, the authors contend that their device holds considerable potential in clean water production and passive salt collection, poised to address global water and environmental challenges. To bolster this claim, I recommend citing specific requirements from internationally recognized contexts, such as the Oman Humanitarian Desalination Challenge.

Reviewer #2 (Remarks to the Author):

As a promising solution to sustainable solar thermal distillation, the authors report a scalable and foldable mangrove-mimicked device for direct solar vapor generation and passive salt collection without brine discharge. Capillarity-driven salty water supply and continuous vapor generation are ensured by anti-fouling porous wicking stem and leaves, which are made of low-cost superhydrophilic nanostructured titanium meshes. Precipitated salt at leaf edge forms porous patch during daytime evaporation and get peeled by gravity during night when saline water rewets the leaves, and these salt patches can enhance vaporization by 1.6 times as indicated by our findings. The proposed solar vapor generator achieves a stable photothermal efficiency. All in all, the topic is interesting. Some comments have been listed as follows:

1. In the previous study, the authors have innovated an artificial tree system that simultaneously acts as a water purifier and a salt crystallizer, which led to a solar-powered desalination technology with zero brine discharge capability. Besides the material, what's the mechanism difference between this topic and that one.
2. The authors proposed a foldable all-in-one mangrove-mimicked solar evaporator by using chemically etched titanium mesh. The advantages of the foldable merits have not been described, and the proofs of foldable merits should be provided.
3. The capillarity-driven salty water supply and continuous vapor generation are ensured in this system, However, in comparison with the more qualitative description is contained, the lack of quantitative description of capillarity-driven salty water supply and continuous vapor generation.
4. Precipitated salt at leaf edge forms porous patch during daytime evaporation and get peeled by gravity during night when saline water rewets the leaves, How to ensure this process occurs repeatedly, since the redissolution of salt requires some environmental conditions.
5. If salt deposits on the surface of SVGC cannot be cleaned in time, the surface blockages would take place, how to prevent it.
6. The authors have mentioned the wicking capability within the SVGC, but the detailed contents have not been found in the manuscript.

Reviewer #3 (Remarks to the Author):

Decision: minor revision

In this manuscript, the authors proposed a scalable and foldable mangrove-mimicked device for direct solar vapor generation and passive salt collection without brine discharge. The biomimetic leaves are very interesting and ingenious, which can be applied in desalination. The porous and crusty salt's effect is clearly explained in the manuscript. Nevertheless, there are still some problems that need to be improved. With a minor revision, the manuscript may be suitable for publication.

1. It is simple to remove the salt off the leaf's edge. However, the accumulation of salt on the surface is possible but unlikely, as shown in figures 4e and S7(a).
2. There are numerous references for combining salt crystallization and evaporation using various architectures. What makes the manuscript novel? In the title and abstract, it is not mentioned.

3. 'Our results have shown that the water supply rate is lower in the shute wires when compared to the warp wires.' It is much better to provide the readers with some possible explanations.
4. 'The sample titled +30° indicated higher thermal efficiency of 75% compared to 60% and 62% for the leaf with 0° and -30° tilt angles, respectively.' It's fascinating. The efficiency of 0°, is the lowest of the three. Can this phenomenon be explained?
5. The unit of the water evaporation flux is $\text{kg}/(\text{m}^2 \cdot \text{h})$. Figure 4d used the produced water weight. The results depended on the leaves' surface area. Please keep the same to remove the influence of the surface area.
6. 'Surprisingly, it can be noticed that the water produced during the dark cycle of the 17 wt% (~2.8 g) is the highest among all salinities as shown in the black columns in Fig.4d.' However, the water produced during the dark cycle of the 3.5 wt% is the highest according to figure 4d. Please check.
7. The durability test is missing since the design is designed to peel the salt passively. Will the performance keep the same after the salt peeling?
8. Will the etched sheet be porous so the droplet leaked instead of spreading according to Figure 1(c)? Several pictures of the spreading process are more convincing.
9. How was the water lifting capacity measured in figure 1(g)?
10. The figures are not clear enough to see the detailed information of the results.
11. There are many references calculating the solar vapor conversion efficiency which are beyond 100%. However, the authors' results are below 100%. Please explain the differences.
12. The authors' excellent work is outstanding. However, the work is not compared with other researchers' work. It is much better to draw a picture to show the improvement.
13. The error bar is missing in the case of Figure 2(d).
14. The width of the stem and structure of the leaves are supposed to affect the water lifting capacity and salt crystallization. Circle structure is supposed to have more effective surface to crystallize and evaporate.
15. In Figure 2(b), the sample of -30° has the highest surface temperature but the evaporation flux is not the highest, please explain the reason.
16. It is doubtful that the velocity of the edge is the same as the surface. Will the accumulated salt not affect the velocity?

Reviewer #1

In this manuscript, the authors introduce a device that integrates a solar vapor generator with a salt crystallizer (SVGC) utilizing edge-preferred crystallization. This device aims to facilitate clean water production and passive salt collection. The authors' findings indicate that the presence of porous and patchy salt within structures resembling clogged leaves can notably enhance the evaporation flux, as compared to completely clean surfaces. Additionally, the authors observe that the decline in evaporation rates and thermal efficiencies of salt-clogged solar evaporators primarily result from losses due to light reflection, rather than pore blockage caused by salt accumulation. In my view, this manuscript merits acceptance for publication, contingent upon the comprehensive addressing of the following comments:

Response:

We are very grateful to the reviewer for recommending our work for publication. The insightful comments have helped us improve the quality of this work significantly. Please check our detailed responses to all the comments given below.

Comment 1: A lucid theoretical exposition of edge-preferred crystallization, ideally rooted in thermodynamics, is currently absent. I strongly encourage the authors to delve into this aspect. Particularly, it is crucial for the authors to expound on the circumstances (pertaining to textile structure, border end-points, etc.) under which crystallization is more prone to occur at the edges.

Response: Thanks for the comment. As suggested, we have provided the theoretical analysis of edge-preferred crystallization in the mesh based on thermodynamics, and we also added the analysis with description to the Main Manuscript and Supporting Information. Please check the details below:

R1. Thermodynamic Analysis of the Edge Crystallization

When using TiO₂/Ti mesh-based solar vapor generator and crystallizer (SVGC), the salt crystallization behavior observed in experiments results from water evaporation, as shown in Figs. R1 and R2. In general, the classical nucleation theory can be used to investigate the change in free energy for homogenous nucleation of salt crystals that is expressed by:¹

$$\Delta G_{\text{hom}} = -\frac{4}{3}\pi r^3 \rho_s \Delta\mu + \pi r^2 \gamma_{lc} \quad (\text{R1})$$

The first term on the right side of Eq. (R1) represents the free energy change due to phase transition, while the second term shows the energy change due to creation of interface between the salt solution and newly forming crystals. r is the radius of the spherical nucleus of new

phase, ρ_s is number density of crystals, $\Delta\mu$ is the difference in chemical potential of solute (i.e., NaCl salt) in solution between supersaturated and saturated states. γ_{lc} is the interfacial tension between liquid phase (i.e., solution) and crystal. In fact, the salt nucleation on titanium mesh surface is heterogeneous. For heterogeneous nucleation on a flat substrate, Eq. (R1) is modified as:¹

$$\Delta G_{het-flat} = -\frac{2}{3}\pi r^3 \rho_s \Delta\mu + \pi r^2 (2\gamma_{lc} + \gamma_{sc} - \gamma_{ls}) \quad (R2)$$

The variables γ_{sc} and γ_{ls} refer to the interfacial tensions between substrate-crystal and liquid-substrate, respectively.

The individual wire of titanium mesh used for our SVGC device is covered with nano/micro-structured titanium dioxide after chemical etching (Fig. 1c in the main manuscript). In addition, the mesh exhibits nearly cone-shaped cavities due to the weaving pattern of wires (Fig. R2d-e). Both the nano/microstructure and cavities will impact the energy barrier for nucleation of salt crystals. The cone angle of the cavities between shute wires illustrated in Fig. R2e is around $\sim 60^\circ$. The liquid bridge between two extended wires (Fig. R2c-d) can also be approximated as cone-shaped cavity though spacing between wires could impact the shape of the liquid bridge. For cone-shaped cavities and heterogeneous nucleation, Eq. R2 can be modified as:²

$$\Delta G_{het-cone} = -\frac{4}{3}\pi r^3 \rho_s \Delta\mu + \pi r^2 \gamma_{lc} \left[\frac{1}{4} \left(2 - 3 \sin \left(\theta + \frac{\psi}{2} \right) + \sin^3 \left(\theta + \frac{\psi}{2} \right) - \cos^3 \left(\theta + \frac{\psi}{2} \right) \cot \left(\frac{\psi}{2} \right) \right) \right] \quad (R3)$$

ψ is the cavity angle while θ is the interfacial angle between solution and nucleated phase as illustrated in Fig. R2e. There are three important parameters in Eq. (R3): $\Delta\mu$, θ and ψ . The impact of these parameters on crystallization behavior of our biomimetic SVGC is analyzed systematically in the subsequent sections.

Figure R1: Optical images for bio-mimicked SVGC after one day cycle (yellow box) and after one night cycle (grey box) using water of various salinities.

Figure R2: Salt precipitation on extended warp wires of SVGC at (a) $t=0$ h and (b) $t=14$ h under ambient conditions. Illustration for the pattern of salt precipitation on the extended warp wires from (c) top and (d) side views. (e) Growth of salt perpendicular to the direction of shute wire and (f) illustration for the salt peeling. The dimensions of the extended warp wires and shute wires are exaggerated for clarity in the schematic given in Fig. (c-f). Figure e also illustrates the nucleation of new phase inside the cone-shaped cavity containing saline water.

R1.1 Effect of Chemical Potential Difference ($\Delta\mu$)

Under low super saturation at constant temperature and pressure, the chemical potential difference for evaporative crystallization of individual species is given by:³

$$\Delta\mu = RT\sigma = RT\left(\frac{c}{c_{eq}} - 1\right) \quad (R4)$$

where R is the universal gas constant, T is the absolute temperature and σ is the relative supersaturation. Similarly, c and c_{eq} are actual and equilibrium (or saturated) salt concentration in solution, respectively. An increase in NaCl concentration will enhance the chemical potential difference of salt in the solution (Eq. R4), thus reducing the energy barrier for salt nucleation (Eq. R3). In our work, the concentration of salt along the mesh length is affected by the evaporation-driven transport of saline water. Therefore, it is important to predict the locations where the concentration of salt exceeds the saturation value and the salt crystallization (i.e., nucleation) occurs most likely. Based on numerical simulation, a detailed analysis of the concentration profile during evaporation along the evaporator length is provided in the following subsection.

R1.1.1 COMSOL Simulation of Variation in Salt Concentration:

In order to study the variation in salt concentration along the stem of SVGC, we carried out the COMSOL simulation for the spatial salt distribution on porous evaporator. The computational domain for the numerical analysis is shown in Fig. R3. A coupled phenomenon of liquid and species transport was modelled by employing the ‘‘Darcy Law’’ and ‘‘Transport of Diluted Species’’ Modules of the COMSOL. The corresponding mass conservation for the liquid transport is written as:

$$\frac{\partial}{\partial t}(\phi\rho) + \nabla \cdot (\rho u) = Q_m \quad (R5)$$

The first term on the left is related to temporal change in mass while the second term represents advective transport of mass driven by evaporation. Q_m is the source term for evaporation (J_v) which is related to the evaporation flux (J_v) and thickness of the mesh (b) as: $Q_m = -J_v / b$. ϕ is the porosity of meshes while ρ is the density of salt water. The liquid transport through the mesh is driven by evaporation and enabled by capillary wicking, which can be modelled through Darcy Law:

$$u = -\frac{K}{\hat{\mu}}(\nabla p - \rho g) \quad (R6)$$

where K is the permeability of meshes ($\sim 60 \mu\text{m}^2$), $\hat{\mu}$ is the viscosity of salt water and p is the pressure. The conservation equation for species transport in saturated porous media can be written as:

$$\phi \frac{\partial c}{\partial t} + \nabla \cdot \left(-\phi \frac{D_e}{\tau} \nabla c \right) + u \cdot \nabla c = S \quad (\text{R7})$$

The first term on the left side of Eq. (R7) represents transient variation in concentration profile while the second term is related to the back diffusion of salt. Similarly, the third term represents the advective transport of salt. τ is the tortuosity of the evaporator while D_e is effective diffusion coefficient of the evaporator. S is the source term for salt concentration and related to evaporation as: $S = -Q_m c / \rho$. Note that Q_m is a negative quantity while S is positive.

Initial and Boundary Conditions:

A constant salt concentration as the inflow boundary condition is applied at Boundary 1 (Fig. R3). At Boundaries 2, 3 and 4, an evaporation flux is applied when considering the contribution from SVGC edges (with free evaporation to the surrounding), and no evaporation flux is considered in the other case without edge effect. As an initial condition, the domain was assumed to be saturated with the concentration of inflow (i.e., bulk solution). The concentration-dependent properties (i.e., density and viscosity) of the saline water are employed in the simulation. A mesh convergence analysis was performed to ensure that results are independent of the grid size.

Assumptions:

1. The back diffusion is neglected owing to the fact that Peclet number is significantly higher than 1 (which means advective transport dominates the diffusive one). Here, the aim is to obtain concentration profile for evaporator with zero liquid discharge rather than salt-resistant evaporator with back diffusion. Most importantly, the steady-state solution for the concentration profile cannot be obtained if the back diffusion is considered.

Thus, only two important competing factors remain to shape the concentration profile: the evaporation-driven advection (transport) of salt ion is countered by the salt generated from precipitation (represented by the source term S), both owing to evaporation. The boundary/location at which both factors are balanced will correspond to $c/c_{\text{sat}}=1$.

2. The effect of gravity is neglected.

It is pertinent to mention that the precipitation process is not a steady-state process but rather a transient one. Though we are making these critical assumptions to obtain a steady-state solution, these assumptions do not affect the accuracy and validity of the simulation results.

3. Evaporation flux is considered uniform over the evaporator surface unless stated.
4. Porous medium (mesh) is homogeneous, and its properties are uniform through the porous matrix.
5. The temperature of the evaporator is uniform.

Figure R3: (a) Computational domain for the COMSOL simulations. (b) Contour map for effect of salinity on salt concentration for two cases i.e., with and without edge effect by using water of various salinities. The black line/curve represents $c/c_{sat} = 1$.

The simulation results for spatial variation of salt concentration along the evaporator length are shown in Fig. R3. It can be seen that the salt concentration increases along evaporator length and reaches the saturation value (i.e., 26.3 wt.% or ~5411 mol/m³) at a certain distance from the inlet (i.e., Boundary 1). As the bulk (or inflow) concentration increases, the length of the probable crystal-free region is reduced. This is applicable to both cases i.e., with and without the edge effect. Figure R4 shows variation in dimensionless salt concentration (c/c_{sat}) as a function of precipitation-free length to the total evaporation length (l_{sat} / L). In the case of no edge effect, the predicted l_{sat} / L values corresponding to $c/c_{sat} = 1$ for 3.5, 6, 12, 18, and 24 wt.% are 0.89, 0.77, 0.6, 0.40, and 0.11, respectively. The l_{sat} / L values become larger when the edge effect is introduced, leading to more crystal-free regions. These simulation results in

Fig. R4b are consistent with the experimental results for salt precipitation with our bio-mimicked SVGC. The deviation between experimental l_{sat} / L (determined based on centerline of the mesh) and simulated l_{sat} / L is due to several reasons. First, the role of back diffusion is neglected in COMSOL simulations. Secondly, the edge of the mesh would provide additional surface for salt precipitation. It should also be noted that crystallization may not necessarily occur at $c/c_{\text{sat}} = 1$ rather at higher values. For instance, nucleation took place at the supersaturation of around $c/c_{\text{sat}} = 1.6$ for the case of glass capillaries filled with NaCl saline solution, as reported by Shahidzadeh et. al.⁴ Nevertheless, the simulation results provide good estimate for the locations where c/c_{sat} is higher than 1 since these locations are more prone to crystallization, owing to the fact that the energy barrier for the nucleation of salt will be reduced at locations where $c/c_{\text{sat}} > 1$ (Eq. R3).

Figure R4: Variation in crystal free length to the total evaporation length (l_{sat} / L) as a function of dimensionless salt concentration (c/c_{sat}). (b) Comparison of simulation results with experimental data for the ratio of salt free length to the total length of evaporator (l_{sat}/L) corresponding to $c/c_{\text{sat}}=1$.

R1.2 Effect of Micro-Structure and Weaving Pattern

Based on Eqs. (R1) and (R3), the ratio of heterogeneous to homogeneous nucleation energy barrier (ΔG^*) can be written as:

$$\Delta G^* = \frac{\Delta G_{\text{het-cone}}}{\Delta G_{\text{hom}}} = \left[\frac{1}{4} \left(2 - 3 \sin \left(\theta + \frac{\psi}{2} \right) + \sin^3 \left(\theta + \frac{\psi}{2} \right) - \cos^3 \left(\theta + \frac{\psi}{2} \right) \cot \left(\frac{\psi}{2} \right) \right) \right] \quad (\text{R10})$$

For heterogeneous nucleation in a conic cavity, the new phase (salt) will form an angle θ with the substrate (mesh wire). Owing to the intricate and complex nature of the nucleation phenomena, θ can vary from 0° to 180° . For heterogeneous nucleation on flat substrate, the cavity angle of $\sim 180^\circ$ is also considered in the analysis. Fig. R5 shows that $\Delta G^* < 1$, which means the energy barrier for heterogeneous nucleation is lower in comparison with

homogeneous one in the whole range of θ . Also, nucleation is more probable to occur in the cavity compared to the flat substrate surface since ΔG^* for $\beta=180^\circ$ is higher than that for $\beta=60^\circ$. Based on the results given in Fig. R5, it can be postulated that the nano/micro structures (as observed in SEM images shown in Fig. 1d in the main manuscript) on the mesh wire will have lower energy barrier compared to the homogeneous case. The overall effect of the low energy barrier, which facilitates nucleation/crystallization, will be the same at all locations of the evaporator, while crystallization is not uniform over the evaporator surface. This clearly demonstrates the dominant impact of chemical potential difference of salt in solution on crystallization, compared with the reduction in nucleation energy barrier due to micro-structure and cavities (quantified through θ and ψ). Moreover, the precipitation behavior observed in experiments is in consistency with the concentration profiles obtained through COMSOL simulation, which confirms the important role of supersaturation.

In fact, some questions remain, for instance, why do we observe quick crystallization at the extended spikes (i.e., wires at Edge-1 in Fig. R2a) compared to non-spike edge of the evaporator (Edge-2 in Fig. R2a)? Is it because of low nucleation energy barrier due to the reduced ΔG^* , supersaturation or something else? As stated before, supersaturation plays a more important role than ΔG^* . Just like the main evaporator surface (where we have l_{sat}/L defined by $c/c_{\text{sat}}=1$ as given in Fig. R3-R4), concentration will also increase along the spike's length and saline water at the tip of spikes will reach the supersaturation state earlier than that at the spike base (Fig. R2a-b). Therefore, higher supersaturation at the spike tip will increase chemical potential difference ($\Delta\mu$) of salt, thus reducing nucleation barrier. However, crystallization behavior will not be same on all spikes, because it also depends on the distance of a particular spike from the evaporator inlet, which affects salt concentration at the base of the spike. Spikes far from the evaporator inlet (boundary 1, Fig. R3a) will have high concentration at their base (as marked in Figs. R2a and R2c), so these spikes will have more salt crystallization in comparison with the ones near the evaporator inlet (as observed in Fig. 5a-b in the main manuscript). In addition, the evaporation rate around the spikes is higher than that on the main evaporator surface (dark region in Fig. R3a) since it is easier for vapor diffusion into air near spikes. Moreover, the liquid film at/between the spikes could be thinner than that at the main surface, and higher evaporation at thin film region will lead saline water to reach supersaturation (c/c_{sat}) more easily, hence more crystallization.

Figure R5: Effect of interfacial angle θ and cone-shape cavity angle (ψ) on dimensionless energy barrier for nucleation.

Comment 2: The fabrication of the proposed textile involves several stages: titanium mesh, oxidation process, and chemical etching. A more intricate characterization of each stage is desirable. For instance, how does the material color evolve during fabrication, from pure titanium to titanium oxide and eventually to nanostructured titanium oxide? Additionally, when referring to titanium oxide, it would be valuable to specify the relevant phase (cubic, anatase, rutile).

Response: As suggested, we have added detailed characterization of nanostructured titanium meshes in the Supporting Information and also summarized as below.

Figure R6 shows the optical images of bare titanium mesh before and after chemical oxidation for 3, 15 and 24 hours. The chemical oxidation time of titanium mesh is critical to control the growth of titanium dioxide nano/microstructures on mesh surface. After chemical oxidation for 3 hours, the color of titanium mesh surface changes from glossy greenish grey to nonglossy dark grey, while the longer oxidation time causes an obvious color change to bluish and whiteish grey. The former indicates the presence of lower concentration/growth of TiO_2 nanostructures, while the latter two cases show much higher concentration/growth of TiO_2 nanostructures at the mesh surface. This is confirmed by surface morphological characterization by using scanning electron microscopy (SEM), as shown in Fig. R6 (b-d). The SEM images show distinguishable TiO_2 nanostructures on both meshes, as lower oxidation time results in submicron particle-like morphology, while the 15 h oxidation time results in

dense grass-like TiO₂ nanowires with diameter below 100 nm. For prolonged oxidation time of 25 h, the grass-like TiO₂ nanowires started to diffuse, showing cluster-like morphology.

Figure R6: (a) Optical images of titanium mesh before and after chemical oxidation for 3, 15, and 24 hours. (b-c) Scanning electron microscopic (SEM) images of respective oxidized TiO₂/Ti mesh.

In addition, we characterized the crystallinity and phases of TiO₂/Ti mesh by using X-ray diffraction (XRD) technique (PANalytical Empyrean). Figure S7 (a-b) shows the XRD pattern of titanium meshes before and after oxidation, which indicates the presence of diffraction peaks corresponding to the TiO₂ nanostructures and Ti substrate. The typical diffraction peak (101) centered at 24.9° indicates the presence of dominant TiO₂ anatase phase. Except for the peaks corresponding to metal titanium, the diffraction peaks of TiO₂/Ti match well the TiO₂ anatase phase peaks with traces of rutile phase, as shown in Fig. S7 (b).^{5,6} The strongest peaks at $2\theta = 24.9^\circ$, 48.0° , and 53.9° are corresponding to the (101), (200) and (105) crystalline planes of anatase phase, while the weak peaks at $2\theta = 61.3^\circ$ and 63.5° represent the (002), (310) crystalline planes of rutile phase, respectively.

Figure R7: (a) XRD spectra of titanium mesh before and after oxidation. TiO₂/Ti mesh after oxidizing for 3 and 24 h, respectively. (b) zoom in of spectra (a) to show the diffraction peaks with lower intensities.

It is worth mentioning that both the effective light absorption and superhydrophilicity for continuous water supply are essential for high-performance SVGC devices. Moreover, we also performed the optical (Perkin Elmer, Lambda 1050) and wettability (Kyowa, contact angle goniometer) characterization. Figure R8 shows the measured absorption spectra of various TiO₂/Ti meshes in the wavelength range of 250-2000 nm. TiO₂/Ti meshes after oxidizing for 3 h exhibit higher light absorption capability than those after oxidizing for 25 h, because TiO₂ nanostructure in the latter case has the higher reflectivity.

Figure R8: Comparison of measured absorption spectra of as-prepared TiO₂/Ti mesh after oxidizing for 3 and 25 h in the wavelength range of 250-2000 nm.

For wettability characterization, we have observed the water droplet spreading behavior on mesh and substrate via high-speed optical imaging. Figure R9 shows the optical images of water droplets in contact with bare titanium and TiO₂/Ti substrates. Bare titanium shows the water contact angle of $66 \pm 2.2^\circ$ after 425 ms, while TiO₂/Ti substrate shows the superhydrophilicity with a water contact angle of $6.3 \pm 0.5^\circ$. Time-lapse optical snapshots of water droplets spreading on bare titanium and TiO₂/Ti meshes are shown in Fig. R10. Water droplets did not wet the bare titanium mesh while instantly spreading on nano/microstructured TiO₂/Ti mesh. Notably, we did not observe any leakage of water across the TiO₂/Ti meshes after 150 ms as shown in Fig. R10.

Figure R9: Optical images of water droplets in contact with (a) bare titanium and (b) TiO_2/Ti substrates.

Figure R10: Time lapse image showing water droplet spreading on (a) bare titanium and (b) oxidized TiO_2/Ti mesh.

Comment 3: The authors allude to the occurrence of back diffusion during nocturnal/dark operation. Have the authors conducted an estimation of the back diffusion coefficient? Does its order of magnitude align with standard self-diffusion values? Alternatively, should a Marangoni effect be taken into consideration? (For reference, see <https://doi.org/10.1039/D0EE01440K>)

Response:

We appreciate this important comment. We have conducted both experimental study and numerical (finite-element) simulation to estimate the back diffusion coefficient of salt (i.e., NaCl) as shown in Fig. R11. The experimental setup along with simulation domain for the estimation of back diffusion coefficient contains single leaf of our SVGC device as shown in Fig. 11. The leaf was dip in saline solution of 3.5 wt.% (Fig. R11a), and the vertical section of the leaf is of 2.2 cm in length while titled one is of 2.4 cm (Fig. R11b). The whole setup was kept in a close chamber to maintain the humidity around ~ 100% to avoid evaporation. A very small droplet of saline water was taken from the evaporator at $z = 22$ mm to measure the salt concentration with refractometer. The experiments were repeated three times to improve the accuracy with a sampling interval of 1 hour.

Fig. R11. Experimental setup for the estimation of back diffusion coefficient with corresponding (b) computational domain for COMSOL simulations. (c) Volumetric concentration profile at $t = 6$ h in

mol/m³, (d) Variation in salt concentration (wt.%) over time at $z = 22$ mm. The solid black line is the simulation data while scatter data points are of experiments.

Transient simulation was performed with COMSOL Multiphysics to investigate the diffusion process and estimate the corresponding diffusion coefficient. A 3D model coupling "Darcy Law" and the "Transport of diluted species" is developed to study the back diffusion process. The dimensions of the computational domain are in accordance with the experiments (Fig. R11b). Saline water (3.5 wt.%) in the container, which is in contact with the base of porous evaporator, was modelled as porous media with porosity of 1 and effective permeability equivalent to a pipe (i.e., $R^2/8$, where R is container radius). Standard value of diffusion coefficient ($\sim 1.5 \times 10^{-9}$ m²/s) was considered for the saline water in container, while the diffusion coefficient for the leaf was adjusted so that the simulated concentration profile matches the experimental data. The Bruggeman model in COMSOL library was employed to calculate the effective diffusion coefficient as a function of tortuosity and porosity. Thermophysical properties of the saline water and governing equations are the same as those in previous simulation (for the response of comment#1). Dirichlet boundary condition is applied at the top edge of the evaporator (Fig. R11b) with fixed concentration of 26.3 wt.%, reproducing the concentration of saline water in-contact with the patchy salt. The leaf is assumed to be initially saturated with 3.5 wt.%. No mass flux boundary conditions ($-n \cdot \rho u = 0$ and $-n \cdot (D \nabla c) = 0$) are applied at all other surfaces except symmetric boundary condition (Fig. R11b) and top edge of the leaf.

Figure R11c shows volumetric contours profile for salt concentration after 6 hr of back diffusion while Fig. R11d reports experimental and simulation results for the variation in salt concentration with time at $z = 24$ mm. By aligning simulation results for purely molecular diffusion with the experimental data, the diffusion coefficient is found to be $\sim 1.15 \times 10^{-8}$ m²/s. The magnitude of the diffusion coefficient is one orders of magnitude higher than the standard diffusion coefficient ($\sim 1.5 \times 10^{-9}$ m²/s). The mechanism of back diffusion could be different from the simple purely molecular diffusion as natural convection and Marangoni effect will induce velocity gradients. The value of back diffusion coefficient ($\sim 1.15 \times 10^{-8}$ m²/s) could have been under-estimated since the circulation of saline water (owing to the natural convection and Marangoni effect) may result in non-uniform concentration profile along the horizontal direction of the evaporator (along x-y plane as shown in Fig. R11b), which could also influence the concentration of saline water at base of the leaf ($z = 0$ mm as shown in Fig. R11b). Surprisingly, we observed a small difference in bulk concentration at $z = 0$ compared to the one

at $z = 22$ mm. This implies the presence of natural convection and Marangoni effect in addition to the purely molecular diffusion. Thus, the actual diffusion coefficient could be higher than the above estimates (i.e., $\sim 1.15 \times 10^{-8} \text{ m}^2/\text{s}$) as reported in.⁷

The surface tension of the saturated saline water (26.3 wt.%, namely $5411 \text{ mol}/\text{m}^3$) is $84 \text{ mN}/\text{m}$, which is 1.15 times of the surface tension of pure water, involving total variation of $\sim 18\%$ ⁸. Similarly, more than 20% variation in density exists between pure water (with density of $997 \text{ kg}/\text{m}^3$) and saturated saline water (with density of $\sim 1200 \text{ kg}/\text{m}^3$).⁹ Therefore, both the density-driven (gravity) and surface tension-driven (i.e., Marangoni effect) flow influence the effective back diffusion coefficient of salt for our SVGC device. The saline water at the top edge of the evaporator (shown in Fig. R11a) will have higher surface tension and higher density in comparison with the saline solution at the base of the evaporator (i.e., bulk solution). A surface tension gradient along the length of the evaporator (owing to concentration gradient) may induce the liquid circulation and assist back diffusion as observed by Morciano et al.¹⁰

It is worth mentioning that our SVGC device has two different types of edges, depending upon the weaving directions (Fig. R2 in our Response of Comment#1, Reviewer#1). The warp wires are parallel to the length of the evaporator (along z direction) while they are perpendicular to the top edge of the evaporator (i.e., perpendicular to x - y plane). These weaving pattern with different edge structure will affect the porosity and tortuosity of the leaf. There will be a small micro-channel of $270 \mu\text{m}$ (which is equivalent to wire diameter) when warp wire illustrated in R2e is removed from the leaf. The porosity of the resulting micro-channel will be almost zero. As effective diffusion coefficient depends on porosity, it will be higher for this specified micro-channel compared to the middle surface of the leaf. The local variations of diffusion coefficient will induce density and surface tension gradient, thus leading to density-driven and Marangoni-driven flow. An extensive investigation on the roles of density-driven flow and Marangoni-driven flow is certainly valuable to understand the mass transfer process during back diffusion. In fact, we believe this is beyond the scope of our current work and has been recommended in our main manuscript for future work.

Comment 4: The authors assert that the proposed material is foldable. I find it somewhat unclear how this foldability offers substantial advantages and ensures optimal light absorption.

Response: Our foldable biomimetic solar evaporator is made of chemically etched titanium meshes. As shown in Fig. R12(a) of Supporting Information, it can retain its shape when folded and bent without fracture. Therefore, we cut and folded the mesh to attain desired shapes such

as small “tree” with four leaves and a big “tree” with many branches at different heights, as shown in Fig. R12(b-c). Mass change comparison for two kinds of devices shows the impact of increasing the number of evaporative leaves (Fig. R12(d)). Besides tree-shape SVGC structure, the mesh was also bent at various title angles to fabricate single-leaf solar vapor generator for studying the effect of tilt angle on evaporation and salt crystallization. Evaporating area changes with the bending angle under the same projection area for light illumination. Owing to the foldable nature of mesh, the leaves of tree-like SVGC device are oriented/folded for maximum exposure to the sunlight during outdoor vapor generation experiments. The results are highlighted in Figure R13 (Fig.2 of the main manuscript). Moreover, our SVGC device can be used as portable freshwater generator in remote areas while saving space compared to bulky devices.

Figure R12: (a) Optical images showing the foldability of titanium mesh used as SVGC. (b) Schematic illustration showing mesh cutting and folding to form tree-like device. (c) Schematic showing top and side view of tree-like structure with different layers and number of leaves (d) Mass change comparison for two tree-like devices.

Figure R13: (a) Optical and IR images of three various setups using saline water with salinity of 12wt% under one sun illumination: single leaf solar vapor generator with tilt angles of -30, 0 and +30°. (b) Comparison of single leaf solar vapor generator with various tilt angles in terms of mass change and evaporation flux per hour.

Comment 5: Lastly, the authors contend that their device holds considerable potential in clean water production and passive salt collection, poised to address global water and environmental challenges. To bolster this claim, I recommend citing specific requirements from internationally recognized contexts, such as the Oman Humanitarian Desalination Challenge.

Response: Thank you for reminding us about the Oman Humanitarian Desalination Challenge (OHDC). We checked the challenge requirements and prepared the following table to demonstrate the capability of our proposed device and potential in passive freshwater production according to the OHDC qualifying criteria as given by the table below:

Requirements	Detailed Requirements	Proposed device potential
Hand-held	The device should be hand-held and easily transportable	The SVGC device is compact, light, foldable, thus suitable for mobile applications
Low-cost	The estimated production cost of the device should be 20 \$	The cost of our SVGC device is about 2 \$ per unit, therefore water productivity can be greatly enhanced if 10 units are used.
Robust	Resilient, corrosion resistant operated through pictorial instructions, long shelf-life, minimal use of easily lost parts	The SVGC device made of titanium dioxide meshes (TiO ₂) is intrinsically anti-corrosive, strong and flexible
Rate of Production	Device should produce a minimum of 3 liters of purified water per day including cloudy days	Our findings show that the device is able to produce 2.2 L/m ² /day when using 4 units in real outdoor conditions. Thus, it is expected to collect over 5 L/day in sunny day when using 10 units, beyond 3 L/day even in cloudy days, at a fixed cost of 20 \$

Stand-alone	There should be no addition of chemicals, fuels, or other external materials, other than the seawater to be purified.	We are using the real seawater directly for freshwater production without any pre-treatment or chemical additives.
Short-term use	The device should operate for a minimum of 30 days.	The proposed device was tested for more than two years since we started this work without considerable efficiency drop
Quality	Device should purify 100 NTU, 35,000 mg/L seawater to 1000 mg/L TDS and meet WHO maximum contaminant levels.	The device is able to purify gulf real seawater with a salinity of 42,000 mg/L to water with a low salinity of 200 mg/L, which meets the WHO requirements well.

Reviewer #2

As a promising solution to sustainable solar thermal distillation, the authors report a scalable and foldable mangrove-mimicked device for direct solar vapor generation and passive salt collection without brine discharge. Capillarity-driven salty water supply and continuous vapor generation are ensured by anti-fouling porous wicking stem and leaves, which are made of low-cost superhydrophilic nanostructured titanium meshes. Precipitated salt at leaf edge forms porous patch during daytime evaporation and get peeled by gravity during night when saline water rewets the leaves, and these salt patches can enhance vaporization by 1.6 times as indicated by our findings. The proposed solar vapor generator achieves stable photothermal efficiency. All in all, the topic is interesting. Some comments have been listed as follows:

Response: We highly appreciate this critical review and insightful comments. We have carefully addressed all of the reviewer's comments and provide detailed responses as below:

Comment 1: In the previous study, the authors have innovated an artificial tree system that simultaneously acts as a water purifier and a salt crystallizer, which leads to a solar-powered desalination technology with zero brine discharge capability. Besides the material, what's the mechanism difference between this topic and that one.

Response: We appreciate this comment and assume the reviewer is referring to the publication below.

Zhang, C., Shi, Y., Shi, L. et al. Designing a next generation solar crystallizer for real seawater brine treatment with zero liquid discharge. *Nature Communications* 12, 998 (2021). <https://doi.org/10.1038/s41467-021-21124-4>

Indeed, this pioneering work focuses on brine treatment with zero liquid discharge, however there are some distinct differences in terms of device design, overall performance and associated mechanism for salt precipitation and shedding. We have summarized these differences in the following bullet points.

- The previous study focused on 3D tetragonal cup shaped structure for concentrated brine treatment with zero liquid discharge, while this work presents a foldable tree-like device for freshwater production and more importantly for passive salt collection, which was not explored before.
- The design in the previous study has two major components: quartz glass fibrous filter membrane and aluminum sheet with spectrally selective cermet coating. Both water evaporation sites and light absorbing surface are physically separated by an aluminum sheet, which could be corroded after long-term operation. In the present work, we

propose a tree-like all-in-one solar vapor generator and crystallizer fabricated by using single material TiO₂/Ti mesh, which has intrinsic properties for anti-corrosion.

- During outdoor operations, 3D tetragonal cup-shaped structure can have maximum light absorption only at noon time, when sunlight irradiates perpendicular to the device. In comparison, our tree-like SVGC device absorbs sunlight from all directions. Owing to the foldable nature, the leaves of the tree can be oriented at different heights for maximum sunlight absorption from sunrise to sunset. Therefore, it can maintain its superior light absorption capability throughout the day.
- 3D tetragonal structure facilitates the salt crystallization exclusively on the outer wall of the crystallizer. The crystallized salt layer could be removed from the wall surface by a plastic spatula or by a mild shock impact. However, our new biomimetic device, consisting of anticorrosive and antifouling nanostructured TiO₂/Ti meshes, supplies salty water via capillary pumping and enables continuous vapor generation until solid salt precipitates at edges of leaves. Salt peeling occurs passively along artificial leaves during night because the precipitated salt patch from leaf edge can be rewetted by saline water to detach and fall under gravity, leading to passive salt collection.
- In our current work, we also revealed how the hydrophilicity of the precipitated salt patches at the edges of the leaves enhances water vaporization by 1.6 times.

The following Table describes the differences briefly.

Previous work	This work
3D tetragonal structure	Foldable porous device
Two different components with absorbing cermet coated aluminum cube and quartz glass fibrous filter membrane	Single component made of anti-corrosive and anti-fouling TiO ₂ /Ti meshes
Directional sunlight absorption	Omnidirectional sunlight absorption
Surface crystallization	Edge crystallization (mechanism of edge-preferred crystallization)
Salt peeling by spatula or mild shock impact	Passive salt peeling at night by gravity

Comment 2: The authors proposed a foldable all-in-one mangrove-mimicked solar evaporator by using chemically etched titanium mesh. The advantages of the foldable merits have not been described, and proofs of foldable merits should be provided.

Response: Our foldable biomimetic solar evaporator is made of chemically etched titanium meshes. As shown in Fig. R14(a) of Supporting Information, it can retain its shape when folded and bent without fracture. Therefore, we cut and folded the mesh to attain desired shapes such as small “tree” with four leaves and a big “tree” with many branches at different heights, as shown in Fig. R14(b-c). Mass change comparison for two kinds of devices shows the impact

of increasing the number of evaporative leaves (Fig. R14(d)). Besides tree-shape SVGC structure, the mesh was also bent at various title angles to fabricate single-leaf solar vapor generator for studying the effect of tilt angle on evaporation and salt crystallization. Evaporating area changes with the bending angle under the same projection area for light illumination. Owing to the foldable nature of mesh, the leaves of tree-like SVGC device are oriented/folded for maximum exposure to the sunlight during outdoor vapor generation experiments. The results are highlighted in Figure R15 (Fig.2 of the main manuscript). Moreover, our SVGC device can be used as portable freshwater generator in remote areas while saving space compared to bulky devices.

Figure R14 (copy of Fig. R12): (a) Optical images showing the foldability of titanium mesh used as SVGC. (b) Schematic illustration showing mesh cutting and folding to form tree-like device. (c) Schematic showing top and side view of tree-like structure with different layers and number of leaves (d) Comparison of mass change comparison for two tree-like devices.

Figure R15 (copy of Fig. R13): (a) Optical and IR images of four various setups using saline water with salinity of 12 wt% under one sun illumination: single leaf solar vapor generator with tilt angles of -30, 0 and +30°. (b) Comparison of single leaf solar vapor generator with various tilt angles in terms of mass change and evaporation flux per hour.

Comment 3: The capillarity-driven salty water supply and continuous vapor generation are ensured in this system, However, in comparison with the more qualitative description is contained, the lack of quantitative description of capillarity-driven salty water supply and continuous vapor generation.

Response: We appreciate this constructive comment and have provided the quantitative analysis of capillarity-driven salty water supply and continuous vapor generation as below: There are two scenarios of wicking phenomena: 1) wicking in dry porous medium without evaporation, 2) wicking driven by evaporation. The wicking velocity is significantly higher under the first scenario (with mm/s as observed in Fig. 1 of the main manuscript), but it becomes very low (\sim in $\mu\text{m/s}$) in Scenario 2. The detailed analysis on wicking capacity of the stem of the SVGC device for Scenario 1 (without evaporation) is provided in our Response to Comment #5 of this Reviewer. And the wicking through the patchy salt has been studied in Fig. 4 in the main manuscript.

On other hand, in solar-driven evaporation (Scenario 2), porous wick is usually saturated by the saline solution before evaporation starts and wicking happens to replenish the amount of liquid for continuous evaporation. This makes the capillarity-driven salty water supply and continuous solar vapor generation a fully coupled phenomenon. In addition, the transport of salt ions from bulk solution to the evaporator surface and precipitation are also coupled with wicking and evaporation.

COMSOL Simulation for Evaporation and Salt Precipitation:

To study the coupled phenomena of wicking, evaporation, and salt precipitation, we carried out the COMSOL simulation for the spatial salt distribution on porous evaporator. The computational domain for the numerical analysis is shown in Fig. R16a. A coupled phenomenon of liquid and species transport was modelled by employing the “Darcy Law” and “Transport of Diluted Species” Modules of the COMSOL. The corresponding mass conservation for the liquid transport is written as:

$$\frac{\partial}{\partial t}(\phi\rho) + \nabla \cdot (\rho u) = Q_m \quad (\text{R11})$$

The first term on the left is related to temporal change in mass while the second term represents advective transport of mass driven by evaporation. Q_m is the source term for evaporation (J_v) which is related to the evaporation flux (J_v) and thickness of the mesh (b) as: $Q_m = -J_v / b$. ϕ is the porosity of meshes while ρ is the density of salt water. The liquid transport through the mesh is driven by evaporation and enabled by capillary wicking, which can be modelled through Darcy Law:

$$u = -\frac{K}{\hat{\mu}}(\nabla p - \rho g) \quad (\text{R12})$$

where K is the permeability of meshes ($\sim 60 \mu\text{m}^2$), $\hat{\mu}$ is the viscosity of salt water and p is the pressure. The conservation equation for species transport in saturated porous media can be written as:

$$\phi \frac{\partial c}{\partial t} + \nabla \cdot \left(-\phi \frac{D_e}{\tau} \nabla c \right) + u \cdot \nabla c = S \quad (\text{R13})$$

The first term on the left side of Eq. (R13) represents transient variation in concentration profile while the second term is related to the back diffusion of salt. Similarly, the third term represents the advective transport of salt. τ is the tortuosity of the evaporator while D_e is effective diffusion coefficient of the evaporator. S is the source term for salt concentration and related to evaporation as: $S = -Q_m c / \rho$. Note that Q_m is a negative quantity while S is positive.

Initial and Boundary Conditions:

A constant salt concentration as the inflow boundary condition is applied at Boundary 1 (Fig. R16). At Boundaries 2, 3 and 4, an evaporation flux is applied when considering the contribution from the SVGC edges (with free evaporation to the surrounding), and no

evaporation flux is considered in the other case without edge effect. As an initial condition, the domain was assumed to be saturated with the concentration of inflow (i.e., bulk solution). The concentration-dependent properties (i.e., density and viscosity) of the saline water are employed in the simulation. A mesh convergence analysis was performed to ensure that results are independent of the grid size.

Assumptions:

1. The back diffusion is neglected owing to the fact that Peclet number is significantly higher than 1 (which means advective transport dominates the diffusive one). Here, the aim is to obtain concentration profile for evaporator with zero liquid discharge rather than salt-resistant evaporator with back diffusion. Most importantly, the steady-state solution for the concentration profile cannot be obtained if the back diffusion is considered.

Thus, only two important competing factors remain to shape the concentration profile: the evaporation-driven advection (transport) of salt ion is countered by the salt generated from precipitation (represented by the source term S), both owing to evaporation. The boundary/location at which both factors are balanced will correspond to $c/c_{\text{sat}}=1$.

2. The effect of gravity is neglected.

It is pertinent to mention that the precipitation process is not a steady-state process but rather a transient one. Though we are making these critical assumptions to obtain a steady-state solution, these assumptions do not affect the accuracy and validity of the simulation results.

3. Evaporation flux is considered uniform over the evaporator surface unless stated.
4. Porous medium (mesh) is homogeneous, and its properties are uniform through the porous matrix.
5. The temperature of the evaporator is uniform.

Figure R16: (a) Computational domain for the COMSOL simulations. (b) Contour map for effect of salinity on salt concentration for two cases i.e., with and without edge effect by using water of various salinities. The black line/curve represents $c/c_{sat} = 1$.

The simulation results for spatial variation of salt concentration along the evaporator length are shown in Fig. R16. It can be seen that the salt concentration increases along evaporator length and reaches the saturation value (i.e., 26.3 wt.% or $\sim 5411 \text{ mol/m}^3$) at a certain distance from the inlet (i.e., Boundary 1). As the bulk (or inflow) concentration increases, the length of the probable crystal-free region is reduced. This is applicable to both cases i.e., with and without the edge effect. Figure R4 shows variation in dimensionless salt concentration (c/c_{sat}) as a function of precipitation-free length to the total evaporation length (l_{sat} / L). In the case of no edge effect, the predicted l_{sat} / L values corresponding to $c/c_{sat} = 1$ for 3.5, 6, 12, 18, and 24 wt.% are 0.89, 0.77, 0.6, 0.40, and 0.11, respectively. The l_{sat} / L values become larger when the edge effect is introduced, leading to more crystal-free regions. These simulation results in Fig. R17b are consistent with the experimental results for salt precipitation with our bio-mimicked SVGC. The deviation between experimental l_{sat} / L (determined based on centerline of the mesh) and simulated l_{sat} / L is due to several reasons. First, the role of back diffusion is neglected in COMSOL simulations. Secondly, the edge of the mesh would provide additional surface for salt precipitation. It should also be noted that crystallization may not necessarily occur at $c/c_{sat} = 1$ rather at higher values. For instance, nucleation took place at the supersaturation of around $c/c_{sat} = 1.6$ for the case of glass capillaries filled with NaCl saline solution, as reported by Shahidzadeh et. Al.⁴ Nevertheless, the simulation results provide good estimate for the locations where c/c_{sat} is higher than 1 since these locations are more prone to

crystallization, owing to the fact that the energy barrier for the nucleation of salt will be reduced at locations where $c/c_{\text{sat}} > 1$.

Figure R17: Variation in crystal free length to the total evaporation length (l_{sat}/L) as a function of dimensionless salt concentration (c/c_{sat}). (b) Comparison of simulation results with experimental data for the ratio of salt free length to the total length of evaporator (l_{sat}/L) corresponding to $c/c_{\text{sat}}=1$.

Comment 4: Precipitated salt at the leave edge forms porous patch during daytime evaporation and get peeled by gravity during night when saline water rewets the leaves. How to ensure this process occurs repeatedly, since the redissolution of salt requires some environmental conditions.

Response: We are grateful to the reviewer for this comment. To demonstrate the repeatability of the edge preferred crystallization throughout the day and salt peeling at night, we performed a new series of uninterrupted indoor experiments for a period of four consecutive days. To mimic the natural day/night alternation, the experimental setup was placed for 12 hours under 1-sun simulated solar irradiance and 12 hours under the dark environment as shown in Fig. R18. In all these experiments, we used water with salinity of 3.5 wt%. Our results have revealed that the mass change curve as function of time is stable and linear during the eight cycles of the experiment: four cycles under light and four cycles in dark (Figure R18(a)). Through the first twelve hours of the experiment when the light was on, the water evaporation flux approximately stayed stable in the range of $0.6 \text{ kg m}^{-2} \text{ h}^{-1}$ as depicted in Fig. R18(b). After twelve-hour operation under solar simulator, a layer of a dense salt crust was formed on the edges of the mangroves shaped structure as shown in Fig. R18(c). It is noteworthy that after turning off the simulated sunlight, the evaporation flux significantly decreased to around $0.2 \text{ kg m}^{-2} \text{ h}^{-1}$ (Figure R18(b)). Moreover, it was noticed that the thick salt layer on the edges of the device was self-defoliated and passively peeled off, resulting in $\sim 0.4 \text{ g}$ salt per day as shown in Fig. R18 (d). Upon switching on the solar simulator again, the evaporation flux was restored,

and the salt patches were crystallized on the edges of the tree. The device was able to obtain a similar evaporation flux in the next operation cycle, indicating that our proposed structure is highly stable and reusable without significant deterioration in the evaporation performance or salt production rate as demonstrated by Fig. R18(b-c).

In our scenario, salt crystallization and accumulation during the day (while the simulated sunlight is on) and automatic passive salt cleaning during the night (under dark environment) will ensure that the device can operate continuously for extended period of time and the solid salt crystals can be peeled regularly without any maintenance or performance degradation.

Figure R18: (a) Indoor experiments: (a-b) measured mass change and evaporation flux for SVGC device consecutive four days. (c) Time laps images for various days showing the device during vapor generation experiments. (d) Mass of collected salt at the end of each day (more salt collection on Day 1 due to dry initial condition).

The mangrove-mimicked device was also tested under real outdoor conditions on the rooftop of Khalifa University in Abu Dhabi, United Arab Emirates (24.45° N and 53.37° E). The measurements were carried out under clear sky for four consecutive days and nights from the morning of 31st of October 2023 to the night of 3rd of November 2023, as presented in Figure R19a below. While performing the outdoor experiments, the average peak direct solar irradiance for four days was 779.5 W m⁻² (blue curve in Figure R19a). The ambient temperature

varies slightly in the range of 28-30 °C over nights and increases during daytime to the peak temperature between 35°C and 37°C (red curve in Figure R19a). On the contrary, the ambient relative humidity was found to be higher at night around 70% and low during the day (green curve in Figure R19a).

These experiments were performed with simulated sea water with a salinity of 3.5 wt %. At the end of the first day (6:30 pm), thick salt layers accumulated on the edge of the evaporator, as shown in Fig. R19b. During the nighttime, the salt layers on the edges of the evaporator were passively peeled and fall down on the insulating foam as dry salt. Similar behaviors of SVGC device were found, that is daytime salt accumulation and nighttime passive salt peeling for consecutive four days. The average mass of passively collected salt for four days is 1.18 g/day. In short, edge crystallization during daytime and passive salt peeling at nighttime was occurring alternatively and uninterrupted until all the bulk water got evaporated, as shown by the time lapse images in Figure R19b.

Figure R19: Outdoor experiments: (a) Measured solar irradiance, ambient temperature, and relative humidity from Oct 31st, 2023, to Nov 03rd, 2023. (b) Time laps images for various days showing the device during vapor generation and salt collection experiments. (c) and (d) measured mass change and evaporation flux for SVGC device from 7:30 am to 9:00 pm on Nov 1st, 2023.

Fig. R19c-d shows the water mass change and evaporation rate over time, recorded on November 1st, 2023. After the sunrise, the mass change was relatively slow as the measured evaporation flux at 8:00 am was 0.55 kg m⁻² h⁻¹. The mass change curve became steeper after 9:00 am and the hourly measured evaporation flux increased linearly until the peak of 2.2 kg m⁻² h⁻¹ at noon, indicating the high influence of solar irradiance in stimulating the solar vapor generation. Following the sunset, the mass change slope became shallow again and the

evaporation flux dropped to about $0.3 \text{ kg m}^{-2} \text{ h}^{-1}$ during nocturnal operation, as presented in Figure R19d. It is worthy to mention that the peak evaporation flux of $2.2 \text{ kg m}^{-2} \text{ h}^{-1}$ during the outdoor experiment was recorded at 780 W m^{-2} (0.78 sun), which is almost four times higher than the indoor evaporation flux of $0.6 \text{ kg m}^{-2} \text{ h}^{-1}$ measured under the irradiance of simulated 1 sun (Fig. 19b). In addition, the average collected mass of salt was 1.18 g/day, almost three times higher than the average collected salt of 0.4 g/day (Fig. R19d) during the indoor controlled experiments. The outdoor experiments indicate the superior impact of heat convection in enhancing both water evaporation and salt production concurrently.

Comment 5: If salt deposits on the surface of SVGC cannot be cleaned in time, surface blockages would take place, how to prevent it.

Response: We appreciate this comment, in fact the salt crystallization is a predictable process. In this work, we have developed a numerical model to predict the location where salt crystallization would take place. The numerical simulation and detailed results are given in our response to Comment #3 of this reviewer. Based on these quantitative predictions, we are able to design appropriate SVGC devices for promoting salt self-peeling and passive salt collection intentionally. Moreover, we have performed experiments and characterized the salt peeling process for the optimal SVGC design for salt self-peeling and passive collection. We also studied the effect of edge structure (i.e., edge with and without spikes) on salt peeling performance. We have found that the sharp spikes on the evaporator edges hinder passive salt peeling while promoting precipitation. Therefore, SVGC without spikes was selected owing to the better salt peeling potential though spikes can favor evaporation. (Please check the detailed study on the effect of spikes in Supporting Information Section 8.1).

Furthermore, we have carried out indoor and outdoor experiments for several consecutive days, which has demonstrated that salt accumulation tends to occur at the edges of our evaporator for 3.5, 7, and 12 wt% salt solutions (as we discussed in our response to the previous comment). Attributed to the continuous wicking during nocturnal operation, the crystalized salt on the edges fell down with the aid of gravity. The residual salt dissolves via back diffusion towards the bulk solution, as we discussed in our response to the previous comment #4). Therefore, we can conclude that the salt peeling performance of our device is desirable for the above-mentioned salt concentrations.

Comment 6: The authors have mentioned the wicking capability within the SVGC, but the detailed contents have not been found in the manuscript.

Response We are thankful to the reviewer for this comment. As reviewer recommended, the detail contents regarding wicking capability of the stem within SVGC are given below:

Propagation of DI water was studied within the as-fabricated stem using infrared imaging as shown in Fig. R20. The propagation distance of the wicking front was monitored with time and results are given in Fig. 1f of the main manuscript. Generally, three forces govern the capillary rate of rise i.e., capillary force is balanced by the gravitational and frictional forces. The capillary pressure (P_c) as a function of surface tension of the working fluid (σ) and pore radius (r_p) and solid-liquid contact angle (θ_{sl}) is governed by the Laplace-Young correlation as given by Eq. (R14):¹¹

$$P_c = \frac{2\sigma \cos \theta_{sl}}{r_p} \quad \text{R14}$$

The pressure due to viscous forces (P_f) depends on the porosity (ϕ) and permeability (K) of the porous material and dynamic viscosity (μ) of the working fluid as given by Eq. (R15).

$$P_f = h \left(\frac{\mu \phi}{K} \right) \left(\frac{dh}{dt} \right) \quad \text{R15}$$

where h of the height of water rise in the titanium sample. It is pertinent to mention that pore radius (r_p) is normally replaced with effective pore radius (r_e) for porous structures by assuming that the solid-liquid contact angle is zero. Combining Eqs. R14 and R15 along with gravitational force, reads as:¹¹

$$\frac{2\sigma}{r_e} = h \left(\frac{\mu \phi}{K} \right) \left(\frac{dh}{dt} \right) + \rho gh \quad \text{R16}$$

where ρ is liquid density, g is gravity, and μ dynamic viscosity. Eq. R16 shows that wicking velocity ($\frac{dh}{dt}$) will decrease with the increase in elevation (h) as confirmed by the experimental results shown in Fig. 1e in the main manuscript. The capillary performance of the wicking materials with complex porous structure (e.g., titanium) are estimated in terms of K/r , which is the ratio of the effective permeability to the effective pore radius.¹² The higher the K/r , the higher the water lifting capacity. Washburn's equation can be employed to determine K/r value as below:¹³

$$h^2 = \left(\frac{4\sigma}{\mu \phi} \right) \left(\frac{K}{r} \right) t \quad \text{R17}$$

This law states that the squared wicking distance will linearly vary with the wicking time. However, this equation does not include the effect of gravity. The curve fitting of the obtained experimental data with the Washburn's equation (for initial few seconds of wicking) provides K/r for the employed wick. The calculated K/r values are $\sim 0.25 \mu\text{m}$ and $\sim 0.5 \mu\text{m}$ for shute wires and warp wires directions, respectively, as shown in Fig. R21. K/r along the warp wire directions is twice than that along the shute wire direction. This implies that the propagation distance for wicking along warp wires will be doubled in comparison with the wicking along shute wires direction.

Figure R20: Vertical wicking heights as a function of time for wicking along warp wires direction. The propagation front is indicated with black dotted lines.

Figure R21: Curve fitted with Lucas–Washburn equation: squared height over time in (a) shute and (b) warp wire direction.

Reviewer #3

In this manuscript, the authors proposed a scalable and foldable mangrove-mimicked device for direct solar vapor generation and passive salt collection without brine discharge. The biomimetic leaves are very interesting and ingenious, which can be applied in desalination. The porous and crusty salt's effect is clearly explained in the manuscript. Nevertheless, there are still some problems that need to be improved. With a minor revision, the manuscript may be suitable for publication.

Response: We really appreciate this positive feedback and valuable comments to help us improve the quality of the work. We carefully considered all of the comments and provided a detailed response to each comment.

Comment 1: It is simple to remove the salt off the leaf's edge. However, the accumulation of salt on the surface is possible but unlikely, as shown in Figures 4e and S7(a).

Response: For solar distillation without liquid discharge, salt precipitates on evaporating surface, and salt accumulation is possible as reported by the early publication (Zhang, C., et al. *Nature Communications* 12, 998, 2021). This is why we propose a new mechanism to promote edge precipitation and remove salt passively in our current work. Though complete salt coverage is unlikely to happen as in Fig. R22 (or Fig. 4 of main manuscript), we are still interested in the extreme case of salt accumulation, where thick salt patches fully cover the evaporator surface. This experiment was performed intentionally to confirm that salt patches can even enhance the evaporation flux for our SVGC device, when compared to the clean device under dark environment. Our findings have indicated that the reduction in evaporation fluxes and thermal efficiencies of the salt-clogged device is mainly due to decrease in absorptivity of SVGC surface rather than blocked pores by the salt. As previously mentioned in the Supporting Information (Fig S16 c-d), the patchy and crusty salt crystals will grow on the mesh surface. By comparing results under different experimental conditions (Figure S16 in the Supporting Information), we observe that the crusty salt crystals are responsible for blocking the pores of the evaporator and reducing the evaporation rate. On the contrary, the patchy salt itself is porous and hydrophilic, which can further enhance the evaporation flux, as shown in Fig R22. In summary, the accumulation of salt on the surface is not random but predictable. More importantly, it can be well controlled to promote passive salt peeling and surface cleaning through the nocturnal operation.

Figure R22: Comparison between clean and salty evaporation devices in terms of mass change under dark environment.

Comment 2: There are numerous references for combining salt crystallization and evaporation using various architectures. What makes the manuscript novel? In the title and abstract, it is not mentioned.

Response: As a promising solution to sustainable solar thermal distillation, our work focuses on developing an innovative foldable mangrove-mimicked device for direct solar vapor generation and passive salt collection without brine discharge. Besides foldable design for SVGC device, the novelty lies in two aspects, as we explored how the crystallized salt at the leaf surface after water evaporation further enhances the evaporation flux, when compared to clean mesh under dark environment. Previously, blocked pores of hydrophilic evaporator were considered to be the major reason for decline in evaporation flux.^{14,15} However, we found that the porous nature of salt crystals promotes efficient and fast water transport in dark environments, thus enhancing the evaporation flux, as shown in Fig. R22. In addition, we concluded that the reduction in evaporation flux is because of the reflection losses and not the blocked pores of the evaporation surface.

- This work describes an innovative foldable device like an artificial tree for freshwater collection while collecting salt as a byproduct.
- Our tree shaped all-in-one SVGC device made from low-cost nano/microstructured TiO₂/Ti meshes supply salty water via capillary pumping and enable continuous vapor generation until solid salt precipitates at edges of leaves.

- Salt peeling can occur passively along artificial leaves during the night because the precipitated salt patch from leaf edge can be rewetted by saline water to detach and fall under gravity, leading to passive salt collection.
- In our current work, we investigated how the hydrophilicity of the precipitated salt patches at the edges of the leaves contribute to increase water vaporization by 1.6 times.

Comment 3: ‘Our results have shown that the water supply rate is lower in the Shute wires when compared to the warp wires.’ It is much better to provide the readers with some possible explanations.

Response

The capillary performance of the wicking materials with complex porous structure are estimated in terms of K/r , which is the ratio of the effective permeability to the effective pore radius. The higher the K/r , higher the water lifting capacity. In warp wires direction, there exists some straight micro-channels with less tortuosity, however, flow path is wavy along shute wires direction. Therefore, the permeability along warp wires is higher compared to shute wires direction. The calculated K/r values are ~ 0.25 and $\sim 0.5 \mu\text{m}$ for shute wires and warp wires directions, respectively. These results are in consistent with the observation of N. Fries et al.¹⁶

Comment 4: The sample titled at $+30^\circ$ indicated higher thermal efficiency of 75% compared to 60% and 62% for the leaf with 0° and -30° tilt angles, respectively.’ It's fascinating. The efficiency of 0° , is the lowest of the three. Can this phenomenon be explained?

Response: Thank you for your comment. The reported thermal efficiency is the average value during the 12-hours experiment. The single leaf evaporator with the 0° tilt angle shows crystallized salt on the center owing to uniform solar heating, as shown in Fig. R23. The accumulated white salt crystals reflect illuminating light instead of absorbing by the mesh, consequently reducing thermal efficiency. Thus, the average solar thermal efficiency of a single leaf evaporator with 0° tilt is the lowest, followed by the one with tilt angle of -30° . In comparison, the single leaf evaporator with a tilt angle of $+30^\circ$ has shown the highest thermal efficiency owing to the fact that the salt crystals tend to precipitate on the edges, leaving the central area clean for efficient light absorption and continuous evaporation through the whole operation period.

Figure R23: Optical and IR images of three various setups using water with salinity of 12 wt% under one sun illumination: single leaf solar vapor generator with tilt angles of -30, 0 and +30°.

Comment 5: The unit of the water evaporation flux is $\text{kg}/(\text{m}^2 \cdot \text{h})$. Figure 4d used the produced water weight. The results depended on the leaves' surface area. Please keep the same to remove the influence of the surface area.

Response: Yes, we agree that the produced water/vapor weight depends on the leaf surface area. Thus, we have carefully revised the manuscript and used the same units " $\text{kg m}^{-2} \text{h}^{-1}$ " throughout the manuscript. We also updated Fig. 4d (main manuscript) after calculating the produced water mass flux in $\text{kg m}^{-2} \text{h}^{-1}$, as shown in Fig. R24 below (Figure 4c in Main Manuscript).

Figure R24: Comparison of produced water and collected salt during the day and night using water of different salinity.

Comment 6: Surprisingly, it can be noticed that the water produced during the dark cycle of the 17 wt% (~2.8 g) is the highest among all salinities as shown in the black columns in Fig.4d.' However, the water produced during the dark cycle of 3.5 wt% is the highest according to figure 4d. Please check.

Response: Thank you for your comment, we have carefully checked all the values and corrected the values in the main manuscript accordingly.

Comment 7. The durability test is missing since the design is designed to peel the salt passively. Will the performance keep the same after the salt peeling?

Response: We are grateful to the reviewer for this comment. To demonstrate the repeatability of the edge preferred crystallization throughout the day and salt peeling at night, we performed a new series of uninterrupted indoor experiments for a period of four consecutive days. To mimic the natural day/night alternation, the experimental setup was placed for 12 hours under 1-sun simulated solar irradiance and 12 hours under the dark environment as shown in Fig. R25. In all these experiments, we used water with salinity of 3.5 wt%. Our results have revealed that the mass change curve as function of time is stable and linear during the eight cycles of the experiment: four cycles under light and four cycles in dark (Figure R25(a)). Through the first twelve hours of the experiment when the light was on, the water evaporation flux approximately stayed stable in the range of $0.6 \text{ kg m}^{-2} \text{ h}^{-1}$ as depicted in Fig. R25(b). After twelve-hour operation under solar simulator, a layer of a dense salt crust was formed on the edges of the mangroves shaped structure as shown in Fig. R25(c). It is noteworthy that after turning off the simulated sunlight, the evaporation flux significantly decreased to around $0.2 \text{ kg m}^{-2} \text{ h}^{-1}$ (Figure R25(b)). Moreover, it was noticed that the thick salt layer on the edges of the device was self-defoliated and passively peeled off, resulting in ~0.4 g salt per day as shown in Fig. R25 (d). Upon switching on the solar simulator again, the evaporation flux was restored, and the salt patches were crystallized on the edges of the tree. The device was able to obtain a similar evaporation flux in the next operation cycle, indicating that our proposed structure is highly stable and reusable without significant deterioration in the evaporation performance or salt production rate as demonstrated by Fig. R25(b-c).

In our scenario, salt crystallization and accumulation during the day (while the simulated sunlight is on) and automatic passive salt cleaning during the night (under dark environment)

will ensure that the device can operate continuously for extended period of time and the solid salt crystals can be peeled regularly without any maintenance or performance degradation.

Figure R25: (a) Indoor experiments: (a-b) measured mass change and evaporation flux for SVGC device consecutive four days. (c) Time laps images for various days showing the device during vapor generation experiments. (d) Mass of collected salt at the end of each day (more salt collection on Day 1 due to dry initial condition).

The mangrove-mimicked device was also tested under real outdoor conditions on the rooftop of Khalifa University in Abu Dhabi, United Arab Emirates (24.45° N and 53.37° E). The measurements were carried out under clear sky for four consecutive days and nights from the morning of 31st of October 2023 to the night of 3rd of November 2023, as presented in Figure R26a below. While performing the outdoor experiments, the average peak direct solar irradiance for four days was 779.5 W m⁻² (blue curve in Figure R26a). The ambient temperature varies slightly in the range of 28-30 °C over nights and increases during daytime to the peak temperature between 35°C and 37°C (red curve in Figure R26a). On the contrary, the ambient relative humidity was found to be higher at night around 70% and low during the day (green curve in Figure R26a).

These experiments were performed with simulated sea water with a salinity of 3.5 wt %. At the end of the first day (6:30 pm), thick salt layers accumulated on the edge of the evaporator,

as shown in Fig. R26b. During the nighttime, the salt layers on the edges of the evaporator were passively peeled and fall down on the insulating foam as dry salt. Similar behaviors of SVGC device were found, that is daytime salt accumulation and nighttime passive salt peeling for consecutive four days. The average mass of passively collected salt for four days is 1.18 g/day. In short, edge crystallization during daytime and passive salt peeling at nighttime was occurring alternatively and uninterrupted until all the bulk water got evaporated, as shown by the time lapse images in Figure R26b.

Figure R26: Outdoor experiments: (a) Measured solar irradiance, ambient temperature, and relative humidity from Oct 31st, 2023, to Nov 03rd, 2023. (b) Time laps images for various days showing the device during vapor generation and salt collection experiments. (c) and (d) measured mass change and evaporation flux for SVGC device from 7:30 am to 9:00 pm on Nov 1st, 2023.

Fig. R 26c-d shows the water mass change and evaporation rate over time, recorded on November 1st, 2023. After the sunrise, the mass change was relatively slow as the measured evaporation flux at 8:00 am was $0.55 \text{ kg m}^{-2} \text{ h}^{-1}$. The mass change curve became steeper after 9:00 am and the hourly measured evaporation flux increased linearly until the peak of $2.2 \text{ kg m}^{-2} \text{ h}^{-1}$ at noon, indicating the high influence of solar irradiance in stimulating the solar vapor generation. Following the sunset, the mass change slope became shallow again and the evaporation flux dropped to about $0.3 \text{ kg m}^{-2} \text{ h}^{-1}$ during nocturnal operation, as presented in Figure R26d. It is worthy to mention that the peak evaporation flux of $2.2 \text{ kg m}^{-2} \text{ h}^{-1}$ during the outdoor experiment was recorded at 780 W m^{-2} (0.78 sun), which is almost four times higher than the indoor evaporation flux of $0.6 \text{ kg m}^{-2} \text{ h}^{-1}$ measured under the irradiance of simulated 1

sun (Fig. 26b). In addition, the average collected mass of salt was 1.18 g/day, almost three times higher than the average collected salt of 0.4 g/day (Fig. R26d) during the indoor controlled experiments. The outdoor experiments indicate the superior impact of heat convection in enhancing both water evaporation and salt production concurrently.

Comments 8: Will the etched sheet be porous, so the droplet leaked instead of spreading according to Figure 1(c)? Several pictures of the spreading process are more convincing.

Response: We would like to thank the reviewer for the constructive comment. For wettability characterization, we have observed the water droplet spreading behavior on mesh and substrate via high-speed optical imaging. Figure R27 shows the optical images of water droplets in contact with bare titanium and TiO₂/Ti substrates. Bare titanium shows the water contact angle of $66\pm 2.2^\circ$ after 425 ms, while TiO₂/Ti substrate shows the superhydrophilicity with a water contact angle of $6.3\pm 0.5^\circ$. Time-lapse optical snapshots of water droplets spreading on bare titanium and TiO₂/Ti meshes are shown in Fig. R28. Water droplets did not wet the bare titanium mesh while instantly spreading on nano/microstructured TiO₂/Ti mesh. Notably, we did not observe any leakage of water across the TiO₂/Ti meshes after 150 ms as shown in Fig. R27.

Figure R27: Optical images of water droplets in contact with (a) bare titanium and (b) TiO₂/Ti substrates.

Figure R28: Time lapse image showing water droplet spreading on (a) bare titanium and (b) oxidized TiO₂/Ti mesh.

Comment 9. How was the water lifting capacity measured in figure 1(g)?

Response: The amount of liquid (\dot{m}) lifted by the stem is calculated as:

$$\dot{m} = \rho g (b w u) \phi \quad \text{R15}$$

where ρ is liquid density (ρ), g is gravity (g) and b is wick thickness, w is stem width (i.e., ~ 40 mm), ϕ is porosity and u is wicking velocity. As the wicking velocity decreases with stem height, the corresponding water lifting capacity will decrease as shown in Fig. R29.

Figure R29: Water lifting capacity of the mimicked stem (with a width of 40 mm equivalent to the proposed device) as a function of height.

Comment 10. The figures are not clear enough to see the detailed information of the results.

Response: We have carefully updated all Figures in the main manuscript as well as in Supporting Information.

Comment 11: There are many references calculating the solar vapor conversion efficiency which are beyond 100%. However, the authors' results are below 100%. Please explain the differences.

Response: The solar vapor generation performance can be evaluated by determining the evaporation efficiency. The evaporation (solar to vapor conversion) efficiency is defined as the ratio of enthalpy difference in the produced vapor (total enthalpy of phase change during the conversion of seawater from liquid to steam) over the total incoming solar irradiance as expressed below:

$$\eta_{th} = \frac{\dot{m}(h_{fg} + C_p \Delta T)}{\alpha q_s A_s} \quad (R16)$$

where \dot{m} represents the transient mass change during the evaporation process in kg s^{-1} , h_{fg} is the enthalpy change from liquid to water vapor $\approx 2400 \text{ kJ kg}^{-1}$, α is the average light absorption, q_s is representing the incident solar flux per area W m^{-2} , and A is the area of the evaporator exposed to the solar irradiations (top illuminated area in our scenario). The energy required for vaporization is divided into two parts, the energy associated with the phase change from liquid to vapor i.e., $\dot{m}h_{fg}$. The second part $\dot{m}C_p\Delta T$ is the sensible heat, where C_p is the specific heat of the saline water. It is noteworthy that the sensible heat is much lower than the latent heat. For instance, for pure water, sensible heat for the temperature gradient of $10 \text{ }^\circ\text{C}$ between the absorber and bulk water (as in the case of our device) is estimated to be 42 kJ kg^{-1} while latent heat of vaporization at $35 \text{ }^\circ\text{C}$ is $h_{fg} = \sim 2422 \text{ kJ kg}^{-1}$. Therefore, the sensible heat is less than 1% of the total useful heat and can be neglected. The evaporation efficiency can be employed as a performance index to compare various solar interfacial desalination devices.

Figure R30: Long 3D solar evaporator (Device 1) and short 2D solar evaporator (Device 2).

Figure R30 shows the comparison of two different devices, made from the same solar absorbing materials. Both evaporators have identical top solar exposed area, therefore for calculating thermal efficiency, we can say that:

$$A_{s1} = A_{s2}$$

However, the mass flux generated from Device 1 will be significantly higher in comparison with the produced vapor from Device 2, as expressed below:

$$\dot{m}_1 > \dot{m}_2$$

Thus, and according to Eq. (R16), the thermal efficiency of device 1 will be higher than the thermal efficiency of device 2 although they both have identical top illuminated area and identical exposed solar light intensity:

$$\eta_{th1} > \eta_{th2}$$

This difference of thermal efficiencies can be explained by considering the temperature profile of two devices, as shown in Fig. R31. Due to the directional solar heating from the vertical solar simulator, the top surface of the solar evaporator will be at higher temperature, leading to generate convective and radiative heat losses to the surrounding environment along with a conductive heat loss to the bulk water beneath. Using an appropriate heat insulation material and constructing a proper water transport path will significantly minimize and block the conductive heat losses.

Dual Thermal Gradient Evaporator

Positive Thermal Gradient Evaporator

Figure R31: Comparison between dual thermal gradient solar evaporator and positive thermal gradient solar evaporator

From a heat transfer point of view, the short and long solar evaporators can be described as positive thermal gradient (PTG) and dual thermal gradient (DTG) evaporators respectively. The working mechanism of the DTG and PTG evaporators is explained in Fig. R31. Under the direct exposure of vertical solar illumination, the long DTG evaporator will encounter combined effect of solar heating on the top illuminated surface and evaporative cooling on the side wall. Due to localized heating by solar energy, a local high temperature will be maintained at the top surface of the long evaporator. As a consequence of heat conduction, a positive thermal gradient $+\left(\frac{\partial T}{\partial y}\right)$ is created on the top area towards bottom side of the long evaporator (left schematic in Fig. R31. This heat conduction from the top surface to the whole solar absorber will not be totally used to evaporate the wicked water. On the other hand, the side wall of the evaporator that is not directly exposed to solar light will be at lower temperature owing to the spontaneous vaporization of water. If the solar evaporator height increased, continuous evaporative cooling will reduce the sidewall temperature to a value less than the bulk water temperature, which will lead to form an upward negative thermal gradient $-\left(\frac{\partial T}{\partial y}\right)$ at the bottom side of the solar evaporator as shown in left schematic of Fig. R31. In this scenario, a low-temperature region will be maintained at the middle part of the long evaporator, which allows the solar absorber to gain energy from the environment and enhance the overall thermal efficiency beyond the theoretical limit. In short, owing to evaporative cooling and environmental heating, the evaporator with dual thermal gradient and long stem will increase the evaporation flux and exhibit extreme high thermal efficiency.

In this work, we fabricated our device with appropriate height since we can take advantage of effective back diffusion through capillary pumped water to facilitate the salt peeling. For further illustration, the back diffusion flux of salt solution can be expressed using Fick's law as:¹⁷

$$J_{bd} = \frac{\rho D(C_s - C_w)}{\delta} \quad \text{R17}$$

where D is the mass diffusion coefficient of salt in water, ρ is the partial density of water in seawater, and C_s and C_w are the mass fractions of salt at the evaporation surface and in bulk solution and δ is the length of mesh where back diffusion is occurring. As we can see from the equation that as the length of mesh is higher, the J_{bd} back diffusion flux will be weaker. This would result in excessive salt accumulation on the surface of mesh, which cannot be peeled on time. Thus, a proper trade-off between high back diffusion and good thermal management is considered in our design for continuous freshwater generation and zero brine discharge simultaneously.

Comment 12: The authors' excellent work is outstanding. However, the work is not compared with other researchers' work. It is much better to draw a picture to show the improvement.

Response: Thanks for the appreciation, we have compared our current work with previous in terms of evaporation flux and thermal efficiency for various materials, as shown in Table below and Fig. R32.

Table R1: Comparison for various materials used as SVG in terms of evaporation flux & thermal efficiency.

Materials	Evaporation Flux	Thermal Efficiency	Hours	Salt Collection	Salinity	Ref
Polyurethane, Polystyrene foam & black paint	-	86	6	No	20	18
Self-assembled aluminum nanoparticles	0.93	57	1	No	2.75	19
Femtosecond laser rendered metal panel	1.26	67	1	No	3.5	20
Fabric wick polystyrene	-	55	0.55	No	3.5	17
Filter paper-CNTs	1.05	81	600	Yes	3.5	21
CuS-coated PE membrane	1.02	63.9	1	No	0	22
Copper-silicon nanowire porous membrane	0.81	50.9	1	No	3.5	23
Ppy-coated Hydrophilic PVDF membrane	0.92	54.3	2.33	no	3.5	24
Electrospun CB/PMMA-PAN Janus absorbers	0.92	51	-	no		25
Janus vertically oriented porous membranes	1.08	62.8	-	no	20	26

Janus SiO ₂ /cellulose nanofiber/carbon nanotube	1.2	80%	100	no	3.5	27
Poly-pyrrol and polyvinyl alcohol (ppy/PVA)	1.09	75	60	Yes	3.5	28
This Work **	1.22	94	12	Yes	3.5	

Figure R32: Comparing the thermal efficiency and evaporation flux of this work to the reported literature.

Comment 13: The error bar is missing in the case of Figure 2(d).

Response: We have repeated the experiments three times for each design, and error bars are added with average evaporation flux and thermal efficiency. Figure 2 of the main manuscript is updated accordingly. In addition, we also added a schematic to show three different designs.

Figure R33: Average evaporation flux and thermal efficiency for single leaf solar vapor generator with various tilt angles.

Comment 14: The width of the stem and structure of the leaves are supposed to affect the water lifting capacity and salt crystallization. Circle structure is supposed to have more effective surface to crystallize and evaporate.

We totally agree with the reviewer about the first point that the width of the stem will certainly affect the water lifting capacity. As previously discussed in the response of comment 9, the amount of liquid (\dot{m}) lifted by the stem is calculated as:

$$\dot{m} = \rho g (bwu) \phi \quad \text{R18}$$

where ρ is liquid density (ρ), g is gravity (g) and b is wick thickness, w is stem width (i.e., ~ 40 mm), ϕ is porosity and u is wicking velocity.

However, regarding the salt crystallization, in our detailed experiments and analysis, we demonstrated that the salt crystallization is dominant and preferred to occur at the edges of the evaporator. Therefore, as more edges are available in the device, more salt crystals will tend to accumulate. As shown in Fig. R34, we are comparing two circular devices, where the first is fully circular and the second is circular with multiple slits (similar to our proposed structure). In the fully circular structure, salt will precipitate in the outer edge, which is the only edge in this case.²⁹

Nevertheless, by cutting the circular structure into semi separated leaves as presented in Fig R34a, more edges will be created for salt crystallization. In addition, one of the main advantages of our proposed device over the previous circular structure is the expandable ability in cascade with other similar devices, forming a multi-layer 3D structure with larger evaporation and salt crystallization areas, as shown in Fig R34b. The two-layer “tree” produced almost double the amount of vapor of 25 g when compared to the 12 g of vapor produced by small tree using 1 sun simulated irradiance with identical illuminating beam area. However, in the case of 2D fully circular structures, the integration of multi evaporators is not practical since the top layer will totally block the incident solar light to reach the bottom layers. Thus, from this comparison, we would like to mention that proper device design and early structure optimization was necessary before fabricating our rational solar evaporator and crystallizer.

Figure R34: (a) Schematic showing two circular devices without and with multiple slits.

Comment 15: In Figure 2(b), the sample of -30° has the highest surface temperature but the evaporation flux is not the highest, please explain the reason.

Response: The temperature shown by IR camera in Fig. 2(b) is measured at the end of the 12-hour experiment to show how salt is accumulated on evaporation single leaf. From Fig. R23, we can clearly see how the salt is accumulating and growing on the edges of leaf with the $+30^\circ$ tilted leaf, which maintains a clean central area. On the contrary, the crusty salt crystals tend to crystallize at the central area on the leaves with tilt angles of 0° and -30° , as shown in time lapse images. This accumulating salt barrier at the leaves surface hinders the water propagation towards the edges and reduces the overall wetted area, consequently resulting in lower evaporation rate. Therefore, the surface temperature of relatively dry areas will be higher in comparison to the wet and clean areas of the leaf. To conclude, the highest surface temperature is owing to the crusty salt accumulation, thus, it will reflect the incident light and generate heat losses to the ambient, causing reduction in the evaporation flux and thermal efficiency.

Comment 16: It is doubtful that the velocity of the edge is the same as the surface. Will the accumulated salt not affect the velocity?

Response:

There are two aspects of wicking phenomena: 1) wicking in dry porous medium without evaporation, 2) wicking driven by evaporation. The edge of the mesh will not play a considerable role during wicking in the former case since wicking velocity is significantly

higher (in mm/s as observed in Fig. 1 given in the main text). In solar-driven evaporation, the wick is usually saturated by the saline solution before evaporation starts (keeping in view Day and Night cycles) and wicking happens to replenish the amount of liquid which evaporates during the process, with very low wicking velocity (\sim in $\mu\text{m s}^{-1}$, Fig. R35). Under these conditions, the edge of the mesh will affect the wicking velocity. To substantiate our claim, we have investigated the velocity distribution during solar-driven evaporation by carrying out COMSOL simulations (Fig. R35). The detailed methodology for the simulations is provided in the response of comment#1 of the reviewer 1.

Results in Figure R35 show that the velocity is uniform along the width of the evaporator if evaporation at the edge of mesh (i.e., edge effect) is neglected, a case similar to the wicking in dry mesh without evaporation. However, when the edge effect is considered, the velocity contours given in Figure R35 shows that the velocity at the edge of the mesh is lower compared to its middle surface at the top end of the evaporator with opposite trend at the base of the evaporator. The wicking velocity during evaporation process, as shown in Fig. R35, is 3 orders of magnitude lower than the one in wicking in dry porous medium (as given in Fig. R35).

Moreover, the accumulated salt will reduce the wicking velocity as it will offer more resistance to flow. Near the vicinity of patchy salt, the concentration of salt will be very high, leading to higher density of the saline water and hence, increase in gravitational force. Our simulation results for salt concentration (consistent with the experiments) shows that concentration is high at the edges of the mesh and precipitation is more prone to occur at edges. Therefore, we will have low velocity at the edge of the mesh. If the salt precipitation is crusty type (i.e., non-porous), they will simply block the flow of the saline solution.

Figure R35: Contours for darcy velocity magnitude ($\mu\text{m s}^{-1}$) and salt concentration for saline water with 3.5 wt.% concentration for two cases: with and without edge effect. Considering edge effect means evaporation at the edge of the mesh is considered. The black line/curve in (b) represents $c/c_{\text{sat}}=1$.

References:

1. De Yoreo, J. J. Principles of Crystal Nucleation and Growth. *Rev Mineral Geochem* **54**, 57–93 (2003).
2. Aili, A., Ge, Q. & Zhang, T. How Nanostructures Affect Water Droplet Nucleation on Superhydrophobic Surfaces. *J Heat Transfer* **139**, (2017).
3. Lewis, A., Seckler, M., Kramer, H. & van Rosmalen, G. *Industrial Crystallization*. (Cambridge University Press, 2015). doi:10.1017/CBO9781107280427.
4. Desarnaud, J., Derluyn, H., Carmeliet, J., Bonn, D. & Shahidzadeh, N. Metastability Limit for the Nucleation of NaCl Crystals in Confinement. *J Phys Chem Lett* **5**, 890–895 (2014).
5. Agrawal, Y. *et al.* High-Performance Stable Field Emission with Ultralow Turn on Voltage from rGO Conformal Coated TiO₂ Nanotubes 3D Arrays. *Sci Rep* **5**, 11612 (2015).
6. Liu, H. *et al.* Sunlight-Sensitive Anti-Fouling Nanostructured TiO₂ coated Cu Meshes for Ultrafast Oily Water Treatment. *Sci Rep* **6**, 25414 (2016).
7. Morciano, M., Fasano, M., Boriskina, S. V., Chiavazzo, E. & Asinari, P. Solar passive distiller with high productivity and Marangoni effect-driven salt rejection. *Energy Environ Sci* **13**, 3646–3655 (2020).
8. Shahidzadeh-Bonn, N., Rafaï, S., Bonn, D. & Wegdam, G. Salt Crystallization during Evaporation: Impact of Interfacial Properties. *Langmuir* **24**, 8599–8605 (2008).
9. Younes, A., Fahs, M. & Ahmed, S. Solving density driven flow problems with efficient spatial discretizations and higher-order time integration methods. *Adv Water Resour* **32**, 340–352 (2009).
10. Morciano, M., Fasano, M., Boriskina, S. V., Chiavazzo, E. & Asinari, P. Solar passive distiller with high productivity and Marangoni effect-driven salt rejection. *Energy Environ Sci* **13**, 3646–3655 (2020).
11. Zhou, F., Zhou, J. & Huai, X. Advancements and challenges in ultra-thin vapor chambers for high-efficiency electronic thermal management: A comprehensive review. *Int J Heat Mass Transf* **214**, 124453 (2023).
12. Jiang, G. *et al.* Ultrathin aluminum wick with dual-scale microgrooves for enhanced capillary performance. *Int J Heat Mass Transf* **190**, 122762 (2022).
13. Jafari, D., Wits, W. W. & Geurts, B. J. Metal 3D-printed wick structures for heat pipe application: Capillary performance analysis. *Appl Therm Eng* **143**, 403–414 (2018).
14. Xu, K., Wang, C., Li, Z., Wu, S. & Wang, J. Salt Mitigation Strategies of Solar-Driven Interfacial Desalination. *Adv Funct Mater* **31**, (2021).
15. Chen, Y. *et al.* Marangoni-driven biomimetic salt secretion evaporator. *Desalination* **548**, 116287 (2023).
16. Fries, N., Odic, K., Conrath, M. & Dreyer, M. The effect of evaporation on the wicking of liquids into a metallic weave. *J Colloid Interface Sci* **321**, 118–129 (2008).
17. Ni, G. *et al.* A salt-rejecting floating solar still for low-cost desalination. *Energy Environ Sci* **11**, 1510–1519 (2018).
18. Zhang, L. *et al.* Highly efficient and salt rejecting solar evaporation via a wick-free confined water layer. *Nat Commun* **13**, 849 (2022).
19. Zhou, L. *et al.* 3D self-assembly of aluminium nanoparticles for plasmon-enhanced solar desalination. *Nat Photonics* **10**, 393–398 (2016).

20. Singh, S. C. *et al.* Solar-trackable super-wicking black metal panel for photothermal water sanitation. *Nat Sustain* **3**, 938–946 (2020).
21. Xia, Y. *et al.* Spatially isolating salt crystallisation from water evaporation for continuous solar steam generation and salt harvesting. *Energy Environ Sci* **12**, 1840–1847 (2019).
22. Shang, M. *et al.* Full-Spectrum Solar-to-Heat Conversion Membrane with Interfacial Plasmonic Heating Ability for High-Efficiency Desalination of Seawater. *ACS Appl Energy Mater* **1**, 56–61 (2018).
23. Song, X. *et al.* Omnidirectional and effective salt-rejecting absorber with rationally designed nanoarchitecture for efficient and durable solar vapour generation. *J Mater Chem A Mater* **6**, 22976–22986 (2018).
24. Wang, Y. *et al.* Improved light-harvesting and thermal management for efficient solar-driven water evaporation using 3D photothermal cones. *J Mater Chem A Mater* **6**, 9874–9881 (2018).
25. Xu, W. *et al.* Flexible and Salt Resistant Janus Absorbers by Electrospinning for Stable and Efficient Solar Desalination. *Adv Energy Mater* **8**, (2018).
26. Yu, H.-H. *et al.* Janus Poly(Vinylidene Fluoride) Membranes with Penetrative Pores for Photothermal Desalination. *Research* **2020**, (2020).
27. Hu, R. *et al.* A Janus evaporator with low tortuosity for long-term solar desalination. *J Mater Chem A Mater* **7**, 15333–15340 (2019).
28. Shao, Y. *et al.* Designing a bioinspired synthetic tree by unidirectional freezing for simultaneous solar steam generation and salt collection. *EcoMat* **2**, (2020).
29. Xia, Y. *et al.* Spatially isolating salt crystallisation from water evaporation for continuous solar steam generation and salt harvesting. *Energy Environ Sci* **12**, 1840–1847 (2019).

REVIEWERS' COMMENTS

Reviewer #1 (Remarks to the Author):

I have carefully read the extensive and detailed rebuttal letter of the authors, as well as the revised manuscript: The authors properly have taken into account my former comments and hence I support the publication of the revised manuscript.

Reviewer #2 (Remarks to the Author):

All the comments from reviewer's have been responded. The manuscript is clear, so I suggest it should be accepted in this revised version.

Reviewer #3 (Remarks to the Author):

The manuscript has been revised as suggested. It is suitable for publication.